# Alkalinity biases in CMIP6 Earth System Models and implications for simulated CO₂ drawdown via artificial alkalinity enhancement

Claudia Hinrichs[1,2], Peter Köhler[1], Christoph Völker[1], Judith Hauck[1]

[1] Alfred-Wegener-Institut Helmholtz-Zentrum für Polar- und Meeresforschung, 27570 Bremerhaven, Germany
[2] now at: Federal Maritime and Hydrographic Agency (BSH), 20359 Hamburg, Germany

*Correspondence to*: Claudia Hinrichs (Claudia.Hinrichs@bsh.de)

**Abstract.** The partitioning of $CO_2$ between atmosphere and ocean depends to a large degree not only on the amount of dissolved inorganic carbon (DIC) but also on alkalinity in the surface ocean. That is also why, in the context of negative emission technologies ocean alkalinity enhancement (OAE) is discussed as one potential approach. Although alkalinity is thus an important variable of the marine carbonate system, little knowledge exists on how its representation in models compares with measurements. We evaluated the large-scale alkalinity distribution in 14 CMIP6 Earth system models (ESMs) against the observational data set GLODAPv2 and show that most models as well as the multi-model-mean underestimate alkalinity at the surface and in the upper ocean and overestimate it in the deeper ocean. The decomposition of the global mean alkalinity biases into contributions from i) physical processes (preformed alkalinity) which includes the physical redistribution of biased alkalinity originating from the soft tissue und carbonates pumps, ii) remineralization, and iii) carbonate formation and dissolution showed that the bias stemming from the physical redistribution of alkalinity is dominant. However, below the upper few hundred meters the bias from carbonate dissolution can become similarly important as physical biases, while the contribution from remineralization processes is negligible. This highlights the critical need for better understanding and quantification of processes driving calcium carbonate dissolution in microenvironments above the saturation horizons, and implementation of these processes into biogeochemical models.

For the application of the models to assess the potential of OAE to increase ocean carbon uptake, a back-of-the-envelope calculation was conducted with each model's global mean surface alkalinity, DIC and $pCO_2$ as input parameters. We evaluate two metrics: 1) the initial $pCO_2$ reduction at the surface ocean after alkalinity addition and 2) the uptake efficiency, $\eta CO_2$, after air-sea equilibration is reached. The relative biases of alkalinity versus DIC at the surface affect the Revelle factor and therefore the initial $pCO_2$ reduction after alkalinity addition. The global mean surface alkalinity bias relative to GLODAPv2 in the different models ranges from -85 mmol m⁻³ (-3.6%) to +50 mmol m⁻³ (+2.1%) (mean: -25 mmol m⁻³ or -1.1%). For DIC the relative bias ranges from -55 mmol m⁻³ (-2.6%) to 53 mmol m⁻³ (+2.5%) (mean: -13 mmol m⁻³ or -0.6%). All but two of the CMIP6 models evaluated here overestimate the Revelle factor at the surface by up to 3.4% and thus overestimate the initial $pCO_2$ reduction after alkalinity addition by up to 13%. The uptake efficiency, $\eta CO_2$, then takes into account that a higher Revelle factor and a higher initial $pCO_2$ reduction after alkalinity addition and equilibration mostly compensate, so that resulting DIC differences in the models are small (-0.1% to 1.1%). The overestimation of the initial $pCO_2$ reduction has to be taken into account when reporting

on efficiencies of ocean alkalinity enhancement experiments using CMIP6 models especially as long as the $CO_2$ equilibrium is not reached.

**Plain text summary**

This study evaluated the alkalinity distribution in 14 climate models and found that most models underestimate alkalinity at the surface and overestimate it in the deeper ocean. It highlights the need for better understanding and quantification of processes driving alkalinity distribution and calcium carbonate dissolution and the importance of accounting for biases in model results when evaluating potential ocean alkalinity enhancement experiments.

**1 Introduction**

Since preindustrial times the ocean has taken up about a quarter of the anthropogenic $CO_2$ emitted into the atmosphere (Friedlingstein et al., 2022). The exact amount of ocean $CO_2$ uptake is determined by the surface ocean carbonate system, which can be largely described by the amount of dissolved inorganic carbon (DIC) and total alkalinity (TA) in the surface ocean (Zeebe and Wolf-Gladrow, 2001). Total Alkalinity is a measure of the excess of bases (proton acceptors) over acids (proton donors) and plays a central role in determining the partitioning of the DIC pool into its

three chemical components, aqueous $CO_2$, bicarbonate ($HCO_3^-$) and carbonate ($CO_3^{2-}$) ions. Aqueous $CO_2$ is the only of the three marine carbonate species that can exchange with the atmosphere. Once in the ocean, most of the additional $CO_2$ taken up is converted into the two other carbonate species. By changing the chemical equilibria between the carbonate species, the ocean carbon uptake leads to ocean acidification with a decrease in pH. This change in the chemical equilibria also reduces the seawater buffer capacity, i.e., the ability of seawater to resist a change in its

carbonate chemistry. The Revelle factor, as a measure of this buffer capacity, is the sensitivity of relative $pCO_2$ (partial pressure of $CO_2$ in seawater) change to relative changes in DIC, and depends both on DIC and TA. A low Revelle factor indicates a high buffering capacity and vice versa (Revelle and Suess, 1957; Middelburg et al., 2020). The lower the Revelle factor, the more DIC occurs as $CO_3^{2-}$ and $pCO_2$ levels in the ocean are lower. This allows the ocean to take up more $CO_2$ which in turn also lowers atmospheric $pCO_2$ (Egleston et al., 2010). Overall, the buffer capacity

implies that the resulting change in pH and $CO_2$ from the same process, e.g., carbonate dissolution, differs depending on the background conditions in TA and DIC (Middleburg et al., 2020). Any changes in pH and $CO_2$ would be smaller in low-sensitivity or well-buffered seawater with a high TA:DIC ratio (low Revelle factor). That is why when Earth System Models (ESMs) are used to quantify the potential $CO_2$ uptake of the ocean, it is important that they simulate reasonable initial states of TA and DIC.

In 2015, the 'Paris Agreement' was adopted by 196 governments at the Conference-of-Parties 21 (COP21). Its goal is to restrict human-induced global warming to well below 2°C, preferably to 1.5°C, compared to preindustrial levels. To accomplish this goal, the signing countries aim to reach peak emissions as quickly as possible and to achieve carbon neutrality by the mid-21st century. This goal is likely not achievable through carbon emission reductions alone according to socio-economic scenario simulations with Integrated Assessment Models (Rogelj et al., 2018). The IPCC

Special Report on Global Warming of 1.5°C states that all (most) projected pathways that limit warming to 1.5°C (2°C) also require use of carbon dioxide removal (CDR) or negative emission technologies (NETs), on the order of 100–1000 Gt $CO_2$ over the 21st century (Rogelj et al., 2018). Existing and potential CDR measures are afforestation and reforestation, land restoration and soil carbon sequestration, bioenergy with carbon capture and storage (BECCS), direct air carbon capture and storage (DACCS), enhanced weathering and ocean alkalinization (Gattuso et al., 2018;

de Coninck et al., 2018; Board and National Academies of Sciences, 2019; National Academies of Sciences, 2021). So far, much research has focused on land-based CDR measures, and it has become clear that it would be extremely difficult to limit global warming to the agreed level with land-based NETs alone (Fuss et al., 2018; Lawrence et al., 2018; Smith et al., 2016).

      Less is known about ocean-based NETs, although some of them appear promising, especially with respect to the

potential scale of application (Gattuso et al., 2018; Boettcher et al., 2019). One promising pathway could be ocean alkalinity enhancement (OAE; Köhler et al., 2013; Renforth and Henderson, 2017). This method is an accelerated version of a natural process: silicate weathering, where alkaline minerals can be mined and crushed (e.g., olivine) or created (e.g., lime) and added to the surface ocean. Alternatively, alkaline solutions from electrochemical weathering can be added. In both scenarios, the alkalinity of the upper ocean is increased and with it the carbon storage capacity

of seawater, which leads to an increased uptake of $CO_2$ from the atmosphere. Aside from lab experiments (Hartmann et al., 2022) and first results from microcosm experiments (Ferderer et al., 2022), these OAE applications are untested at larger scales, so that simulations with state-of-the-art ESMs are essential for assessing the efficiency and biogeochemical implications of ocean alkalinization. Previous model experiments have provided first estimates of the efficiency for idealized experiment set-ups (e.g., Ilyina et al., 2013; Köhler et al., 2013; Keller et al., 2014; Hauck et

al., 2016; González and Ilyina, 2016; Lenton et al.,2018; Burt et al.,2021. Although these modeling studies have suggested that OAE may be a viable method to help reduce atmospheric $CO_2$, the results are difficult to compare due to different experimental designs. Another caveat is that previous estimates of OAE efficiency and side effects were based on single model experiments and did not include a thorough assessment of simulated alkalinity and model-dependence of the results. Now, more and more projects are underway or in planning that seek to apply more realistic

scenarios for OAE e.g., in regional OAE applications (Butenschön et al.,2021; Wang et al., 2023) or coastal applications (Feng et al., 2017; He and Tyka, 2023), which is why a model evaluation is even more important. Furthermore, the development of standards for monitoring, reporting and verification (MRV) methods for real-world OAE applications is currently underway and it becomes clear, that because of the complexity of the carbonate system and the insufficient maturity of observational sensors, numerical simulations are required to fulfill these MRV

requirements (Ho et al., 2023; Bach et al, 2023). Therefore, the continuous development of suitable, carefully validated models is a critical part of this effort (Ho et al., 2023).

      There have been a number of studies that evaluate the simulation of ocean biogeochemical parameters in state-of-the-art ESMs that contributed to CMIP6, the 6[th] phase of the Coupled Model Intercomparison Project (Eyring et al., 2016), but did not include the evaluation of alkalinity (Séférian et al., 2020;, Tagliabue et al., 2021; Kwiatkowski et al.,2020)

or if so then only with one global score number (Terhaar et al., 2022; Fu et al., 2022). The recent study by Planchat et

al. (2023) assessed simulated alkalinity and parameters related to the carbonate pump in CMIP6 models and their predecessor CMIP5 versions. They report an improvement in the representation of alkalinity and the carbonate pump in CMIP6 versus CMIP5. While some models did increase in complexity, they find that potential effects of future ocean changes (e.g., ocean acidification) are not well constrained in many models.

Here, we present further analyses of biases in alkalinity and DIC in CMIP6 models. We show how those biases can be attributed to the ocean's physical, soft-tissue, or carbonate counter pump following Koeve et al. (2014). Furthermore, we provide an estimate of each model's carbonate system sensitivity to OAE depending on their alkalinity and DIC bias in historical simulations.

## 2. Methods

**2.1. CMIP6 models and observational data products**

Our evaluation includes 14 ESMs with ocean biogeochemistry modules from ten modelling centers that contributed to CMIP6 and that provided the variables *dissic* (DIC [mol m$^{-3}$]), *no3* (nitrate concentration [mol m$^{-3}$]), *o2* (dissolved oxygen concentration [mol m$^{-3}$]), *ph* (seawater pH on total scale), *po4* (phosphate concentration [mol m$^{-3}$]), *so* (salinity (S) [g kg$^{-1}$]), *talk* (TA [mol m$^{-3}$]), and *thetao* (potential temperature [°C]), Table 1).

*Table 1: Overview of CMIP6 models considered in this study listing the climate model name and description paper, the model ocean component, the model biogeochemistry component, horizontal grid resolution, number of vertical levels, the ensemble member considered and the data reference*

| CMIP6 ESM | Ocean Model | Ocean Biochem. Model | Ocean Horizontal Resolution (lon x lat) | Ocean vertical levels | Member / Dataset Reference |
|---|---|---|---|---|---|
| ACCESS-ESM-1.5 (Ziehn et al., 2020) | MOM5 | WOMBAT | 360 x 300 (tripolar, ~1°) | 50 | r1i1p1f1 (Ziehn et al., 2019) |
| CanESM5 (Swart et al., 2019b) | NEMO3.4 | CMOC | 361 x 290 (tripolar, ~1°) | 45 | r1i1p1f1 (Swart et al., 2019a) |
| CESM2 (Danabasoglu et al., 2020) | POP2 | MARBL | 320 x 384 (~1°) | 60 | r1i1p1f1 (Danabasoglu, 2019a) |
| CESM2-WACCM (Danabasoglu et al., 2020) | POP2 | MARBL | 320 x 384 (~1°) | 60 | r1i1p1f1 (Danabasoglu, 2019b) |
| CNRM-ESM2-1 (Séférian et al., 2019) | NEMO3.6 | PISCESv2-gas | 362 x 294 (tripolar, ~1°) | 75 | r1i1p1f2 (Seferian, 2018) |
| GFDL-CM4 (Held et al., 2019; Dunne et al., 2020a) | MOM6 | GFDL-BLINGv2 | 1440 x 1080 (tripolar, ~ 0.25°) | 75 | r1i1p1f1 (Guo et al., 2018) |
| GFDL-ESM4 (Dunne et al., 2020b) | MOM6 | GFDL-COBALTv2 | 720 x 576 (tripolar, ~0.5°) | 75 | r1i1p1f1 (Krasting et al., 2018) |
| IPSL-CM6A-LR (Boucher et al., 2020) | NEMO-OPA | PISCESv2 | 362 x 332 (tripolar, ~1°) | 75 | r1i1p1f1 (Boucher et al., 2018) |
| MPI-ESM1-2-HR (Müller et al., 2018; Mauritsen et al., 2019) | MPIOM1.63 | HAMOCC6 | 802 x 404 (~0.4°) | 40 | r1i1p1f1 (Jungclaus et al., 2019) |
| MPI-ESM1-2-LR (Mauritsen et al., 2019) | MPIOM1.63 | HAMOCC6 | 256 x 220 (~1.5°) | 40 | r1i1p1f1 (Wieners et al., 2019) |
| MRI-ESM2-0 (Yukimoto et al., 2019a) | MRI.COM4.4 | MRI.COM4.4 | 360 x 364 (tripolar, ~1°) | 61 | r1i2p1f1 (Yukimoto et al., |

| | | | | | 2019b) |
|---|---|---|---|---|---|
| NorESM2-LM (Tjiputra et al., 2020) | MICOM | iHAMOCC | 360 x 384 (~1°) | 70 | r2i1p1f1 (Seland et al., 2019) |
| NorESM2-MM (Tjiputra et al., 2020) | MICOM | iHAMOCC | 360 x 384 (~1°) | 70 | r2i1p1f1 (Bentsen et al., 2019) |
| UKESM1-0-LL (Sellar et al., 2019) | NEMO-HadGEM3-GO6.0 (eORCA1) | MEDUSA2 | 360 x 330 (tripolar,~1°) | 75 | r1i1p1f2 (Tang et al., 2019) |

For the 14 CMIP6 ESMs, monthly data from one ensemble member (see Table 1) of the historical simulation was downloaded from the CMIP6 archive (https://esgf-data.dkrz.de), post-processed and regridded with bilinear remapping onto a common 1°x1° grid using Climate Data Operators (cdo, Schulzweida, 2022). TA is often normalized ($TA_n$) with salinity to exclude the freshwater effect in the alkalinity assessment (Millero et al., 1998; Fry et al., 2015). Salinity normalization of alkalinity was achieved by using a reference salinity of 35 g kg$^{-1}$. Grid points with a salinity smaller than 10 were masked to avoid very high $TA_n$ values, e.g., from the Baltic Sea:

$$TA_n = \frac{TA}{S} \times 35,  \tag{1}$$

with S being the grid point salinity. The present-day (1995-2014) model climatologies from the historical simulations are evaluated against gridded observational products: (i) TA, DIC and pH from the GLODAPv2.2016b Mapped Climatology (in the following GLODAP, Lauvset et al., 2016); (ii) oxygen and nutrients from the World Ocean Atlas 2018 dataset (WOA, Garcia H.E., 2019) and GLODAP; and (iii) salinity and temperature from the Polar science center Hydrographic Climatology (PHC3.0, Steele et al., 2001) and WOA. For the evaluation of global mean vertical profiles, the model data are interpolated onto the same 33 vertical levels used in the GLODAP climatology. For the purpose of model assessment, the GLODAP TA and DIC data are converted from units of µmol kg$^{-1}$ to mmol m$^{-3}$ using the potential density computed from GLODAP salinity and temperature data.

**2.2. Analysis of the vertical distribution of total alkalinity – the TA* method**

In order to better understand the vertical distribution of modeled alkalinity compared to the observed one, we follow the 'TA* method' as described by Koeve et al. (2014). This method aims to separate the effects of biogeochemical processes and ocean circulation on the distribution of TA. To achieve this, TA is separated into three components: preformed TA ($TA^0$), TA decrease from remineralization of organic matter ($TA^r$), and TA increase due to calcium carbonate ($CaCO_3$) formation and dissolution (TA*):

$$TA = TA^0 + TA^* - TA^r \ [\text{mmol m}^{-3}]  \tag{2}$$

Preformed TA represents the TA of a water parcel when it was last in contact with the atmosphere. This preformed TA is derived by applying multi-linear regression of upper ocean (here top 100 m) salinity, potential temperature, and PO (a conservative water-mass tracer analog to NO in Broecker (1974)) for each model, where

$$PO = O_2 + r_{-O2:PO4} \cdot PO_4,  \tag{3}$$

with $r_{-O2:PO4} = 170$, onto upper ocean TA values (Koeve et al., 2014). The obtained regression coefficients are then applied to salinity, potential temperature, and PO everywhere in the interior ocean to compute the model's $TA^0$ at any location. This preformed alkalinity also includes the physical redistribution of alkalinity biases stemming originally from soft tissue and carbonate pumps and the upwelling of water masses with biased alkalinity.

The $TA^r$ term describes the reduction of TA stemming from the remineralization of organic matter. This term can be described as a function of the simulated Apparent Oxygen Utilization (AOU, Garcia and Levitus, 2006):

$$TA^r = r_{Alk:NO3} \cdot r_{NO3:-O2} \cdot AOU, \qquad (4)$$

with $r_{Alk:NO3}$ = 1.26, $r_{NO3:-O2}$ = 1/10.625 (Koeve et al., 2014), and AOU as difference between oxygen saturation computed following Weiss (1970) and oxygen concentration $O_2$.

Lastly, the contribution from carbonate formation and dissolution, TA*, is computed as residual after rearranging Eq. (2).

We applied the TA* method to 10 of 14 CMIP6 ESMs (CNRM-ESM2-1, GFDL-CM4, GFDL-ESM4, IPSL-CM6A-LR, MPI-ESM1-2-HR, MPI-ESM1-2-LR, MRI-ESM2-0, NorESM2-LM, NorESM2-MM and UKESM1-0-LL), which had the necessary output fields (*talk*, *so*, *thetao*, *o2*, and *po4*).

### 2.3. Theoretical Model Sensitivity to Alkalinity Enhancement

Systematic biases in TA and DIC have implications for a model's theoretical carbonate system sensitivity to added alkalinity during OAE. Thus, differences in ocean carbon uptake and pH increase may occur. In order to evaluate the range of this carbonate system sensitivity we conducted back-of-the-envelope-calculations for all ESMs and the GLODAP dataset using the Matlab toolbox CO2SYS (Lewis et al., 1998; Van Heuven et al., 2011). This toolbox, from any combination of two out of five variables (DIC, TA, pH, $pCO_2$, $fCO_2$), computes the values of the missing variables and derived quantities. Here, we use the time and area-weighted mean surface TA and DIC (Figure 1), converted from mmol $m^{-3}$ to µmol $kg^{-1}$ with a density of 1,026 kg $m^{-3}$, see Table S1 for input values. Additionally, we use the following values for the computation of the carbonate systems: salinity = 34.0, potential temperature = 15 °C, silicic acid = 2 µmol $kg^{-1}$, and phosphate = 1 µmol $kg^{-1}$. Gas exchange with the atmosphere is not considered in any of our theoretical calculations. First, we evaluate the CO2SYS output fields Revelle factor, pH, and $pCO_2$ for the CMIP6 ESMs against the values for the GLODAP data. In a second step, we assess the initial changes in surface $pCO_2$ and pH after an addition of 100 µmol $kg^{-1}$ TA (corresponds to 102.6 mmol $m^{-3}$ TA) while keeping DIC constant. In a third step, we evaluate the $CO_2$ uptake efficiency ($\eta CO_2$) (Renforth and Henderson, 2017, Tyka et al., 2022) and the pH difference at constant $pCO_2$ which simulates completed air-sea $CO_2$ equilibration. Note, that the latter calculation has an ocean-centric perspective as it assumes constant atmospheric $CO_2$, which contradicts the motivation for OAE to reduce atmospheric $CO_2$, and thus will only be valid for small-scale applications. The uptake efficiency metric has been previously applied in ocean model simulations with constant and non-interactive atmospheric $CO_2$ (Tyka et al., 2022; He and Tyka, 2023). We here follow this approach in our idealized calculations while acknowledging that atmospheric $CO_2$ would drop in emission-driven simulations (magnitude dependent on amount of alkalinity added;

Ferrer Gonzalez et al., 2018; Lenton et al., 2018; Köhler, 2020), as in the real world, through feedbacks with the atmosphere and the land biosphere (Oschlies, 2009). The assumption of constant atmospheric $CO_2$ (and thus constant surface ocean $pCO_2$) was shown to overestimate oceanic $CO_2$ uptake by 2% on annual timescale, but by 25% on decadal timescale and further increasing on longer timescales (Oschlies, 2009).

The uptake efficiency, $\eta CO_2$, is the ratio of moles of $CO_2$ absorbed to moles of added alkalinity and can also be expressed as the ratio of the partial pressure sensitivities of $pCO_2$ with respect to TA and to DIC (Tyka et al., 2022; He and Tyka, 2023):

$$\eta CO_2 = \frac{\Delta DIC}{\Delta TA} = - \frac{\frac{\partial pCO_2}{\partial TA}}{\frac{\partial pCO_2}{\partial DIC}} \tag{5}$$

For the uptake efficiency at constant $pCO_2$, the $\Delta DIC$ was also computed using CO2SYS, here with TA + 100 µmol kg$^{-1}$ and the initial $pCO_2$ as input parameters.

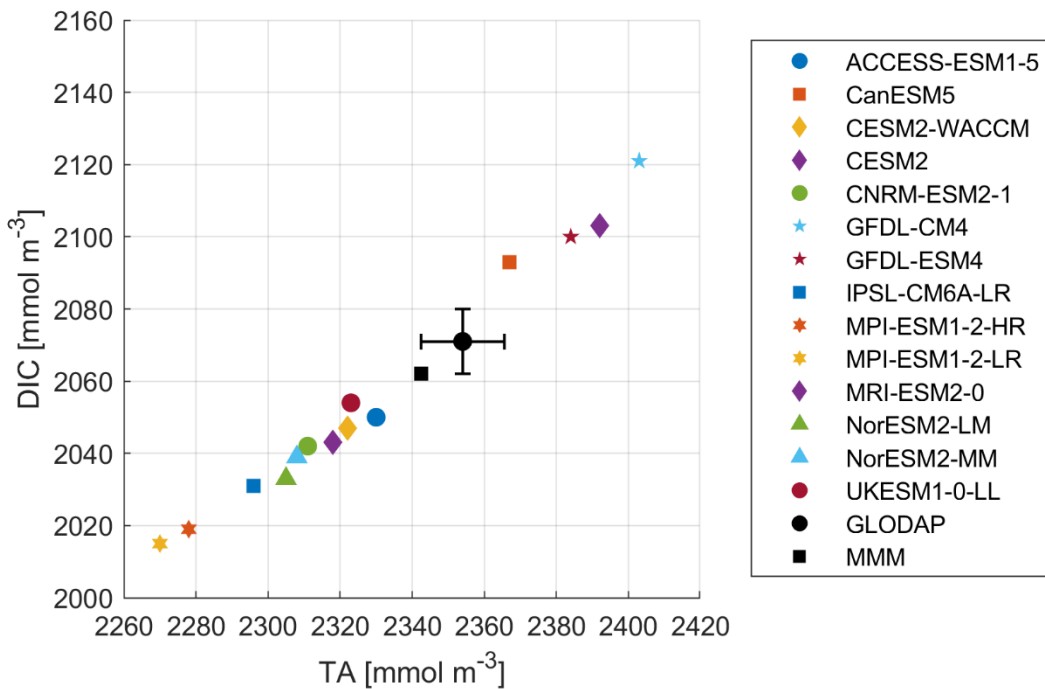

**Figure 1: Global mean surface DIC [mmol m$^{-3}$] versus TA [mmol m$^{-3}$] of the 14 CMIP6 ESMs, the multi-model-mean (MMM), and GLODAP including its error estimate.**

## 3. Results

### 3.1. Analysis of CMIP6 alkalinity and DIC

The comparison of the models' simulated TA at the ocean surface to the GLODAP climatology shows that – on a global scale - most models underestimate surface TA and DIC, except for four models, CanESM5, GFDL-CM4, GFDL-ESM4 and MRI-ESM2-0, which simulate too much TA and DIC at the surface (Figures 1, 2). The multi-model-mean (MMM) is only slightly negatively biased (Figures 1, 2). Global mean surface TA and DIC biases are strongly correlated (R=0.99, Figure 1). Near-surface TA is strongly correlated with salinity, and upper ocean salinity is

governed by freshwater fluxes, e.g., precipitation and evaporation (Millero et al., 1998), and river flows (Cai et al., 2010). Overall, the comparison of salinity-normalized $TA_n$ to GLODAP data shows bias patterns very similar to those of TA for all models. Most notably, some regional peculiarities that stem from salinity biases rather than biogeochemical processes are smoothed out (e.g., North Atlantic bias in NorESM) (Figure S1).

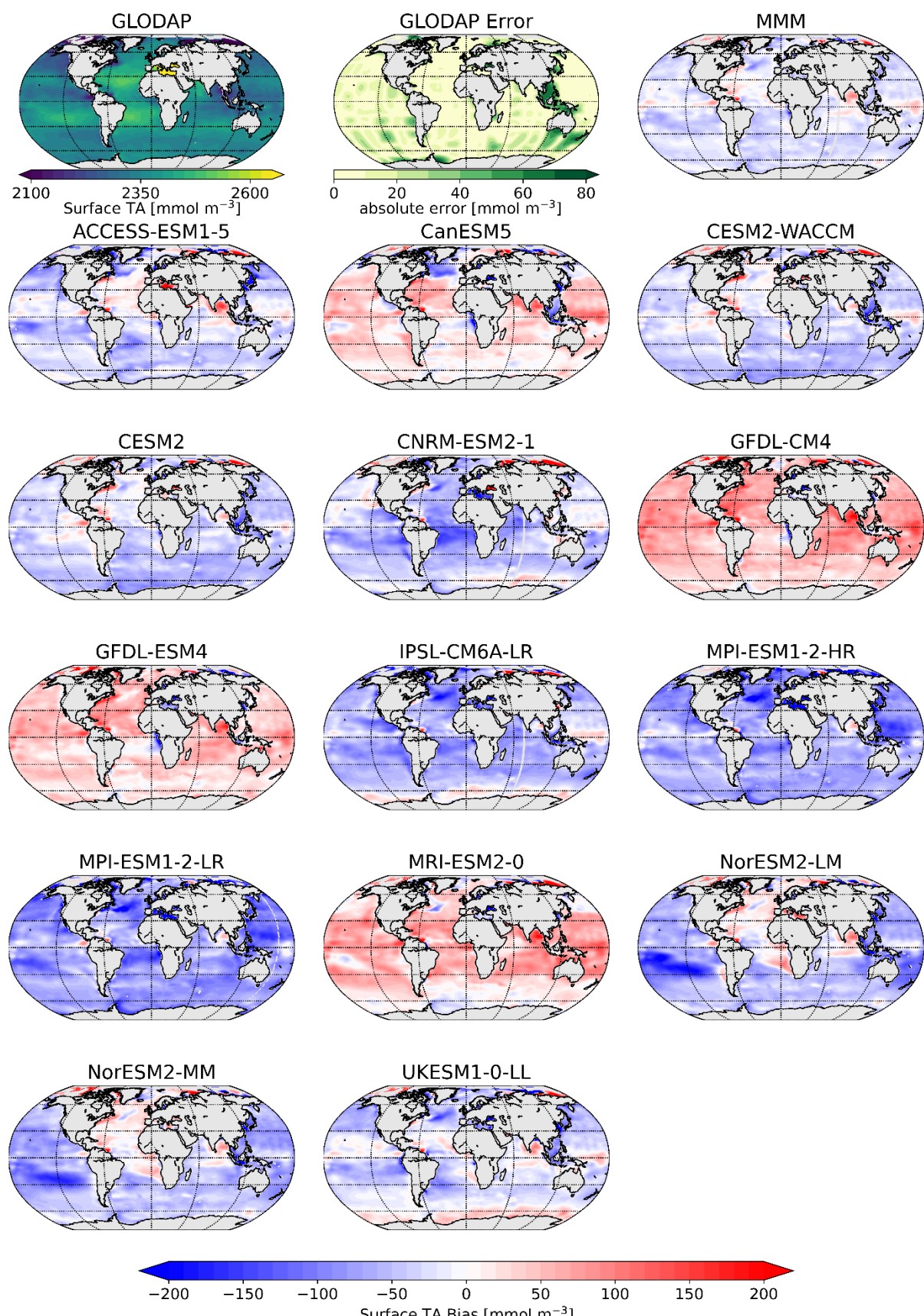


**Figure 2: Surface distribution of TA in GLODAP (top left), its error estimate (top center) and the CMIP6 multi-model-mean (MMM) bias (top right), as well as the respective biases of the ESMs.**

The vertical profiles of globally averaged TA and normalized $TA_n$ (Figure 3) show the aforementioned distribution of the CMIP6 models' surface bias as well, with most of the models showing less surface TA than GLODAP. The models

mostly reproduce the features of the observed TA depth profile: the surface minimum, the subsurface maximum of TA, another minimum at around 500 m depth and the increase of TA with deeper depth (Figure 3a). Two models of the same family (MPI-ESM1-2-LR and MPI-ESM1-2-HR) have less TA than the GLODAP product over the whole water column and two models (GFDL-CM4 and GFDL-ESM4) have higher TA overall. This indicates that their global inventory of TA is too low (too high) compared to GLODAP. The explanation for the systematic low bias in the MPI

model seems to be that too much TA was lost to the sediments during the model spin up (Koeve et al., 2014; Planchat et al., 2023). The high TA bias in the GFDL ESMs was apparently introduced in the post-processing step during the unit conversion from gravimetric ($\mu$mol kg$^{-1}$) to volumetric (mmol m$^{-3}$, common SI unit). The unit conversion is usually based on a chosen density value which is not prescribed in modeling protocols. While most models chose a value between 1,024 kg m$^{-3}$ and 1,028 kg m$^{-3}$, the modeling group at GFDL apparently converted the units using a

value of 1,035 kg m$^{-3}$ (Planchat et al., 2023). The profiles of the other models show either too little TA at the surface and too much at depth, or vice versa, indicating that their TA inventory is closer to the observed one but that the TA distribution in the water column differs from the observations. Salinity-normalization generally does not change the bias patterns (Figure 3b). The salinity-normalization does affect the shape of the profiles in the upper ocean. The surface minima and the subsurface maxima seen in TA disappear. Those features are essentially related to the upper

ocean salinity distribution.

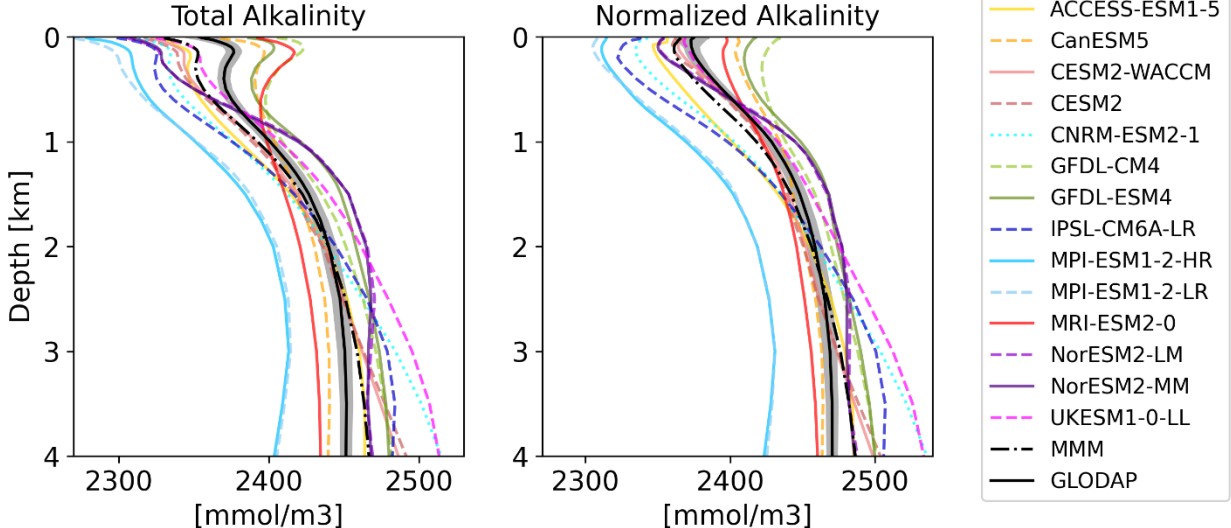

**Figure 3: Vertical profiles of global mean TA (a) and $TA_n$ (b) of the CMIP6 ESMs, the multi-model-mean (MMM) and GLODAP (black) with error estimate (grey shading)**

The near-surface TA maximum seen in the global profile is also evident in the Atlantic, Pacific and Indian Oceans (Figure 4). The high TA is related to the salinity maxima of subtropical underwater in the respective basins (Talley, 2002) and all models replicate this pattern. In the Atlantic Ocean, a TA minimum can be observed in the GLODAP data at around 800 m depth which represents Antarctic Intermediate Water in the South Atlantic (low salinity) (Takahashi et al., 1981). This minimum is not well reproduced by the ESMs, referring to circulation biases. The

relatively low TA in the deep Atlantic Ocean (compared to the Pacific and Indian Ocean), between 1,500 m and 3,500 m depth, and the small gradient with depth is linked to North Atlantic Deep Water (NADW). Most models reproduce this pattern, while the CNRM, IPSL and UK ESMs simulate a strong increase of TA below about 2,000 m depth (Figure 4b). Those three ESMs have a NEMO ocean model in common. The profile shapes in the Southern Ocean and Arctic Ocean are generally reproduced in terms of the TA gradients with depths, albeit the biases in absolute amount

of TA present are visible here as well.

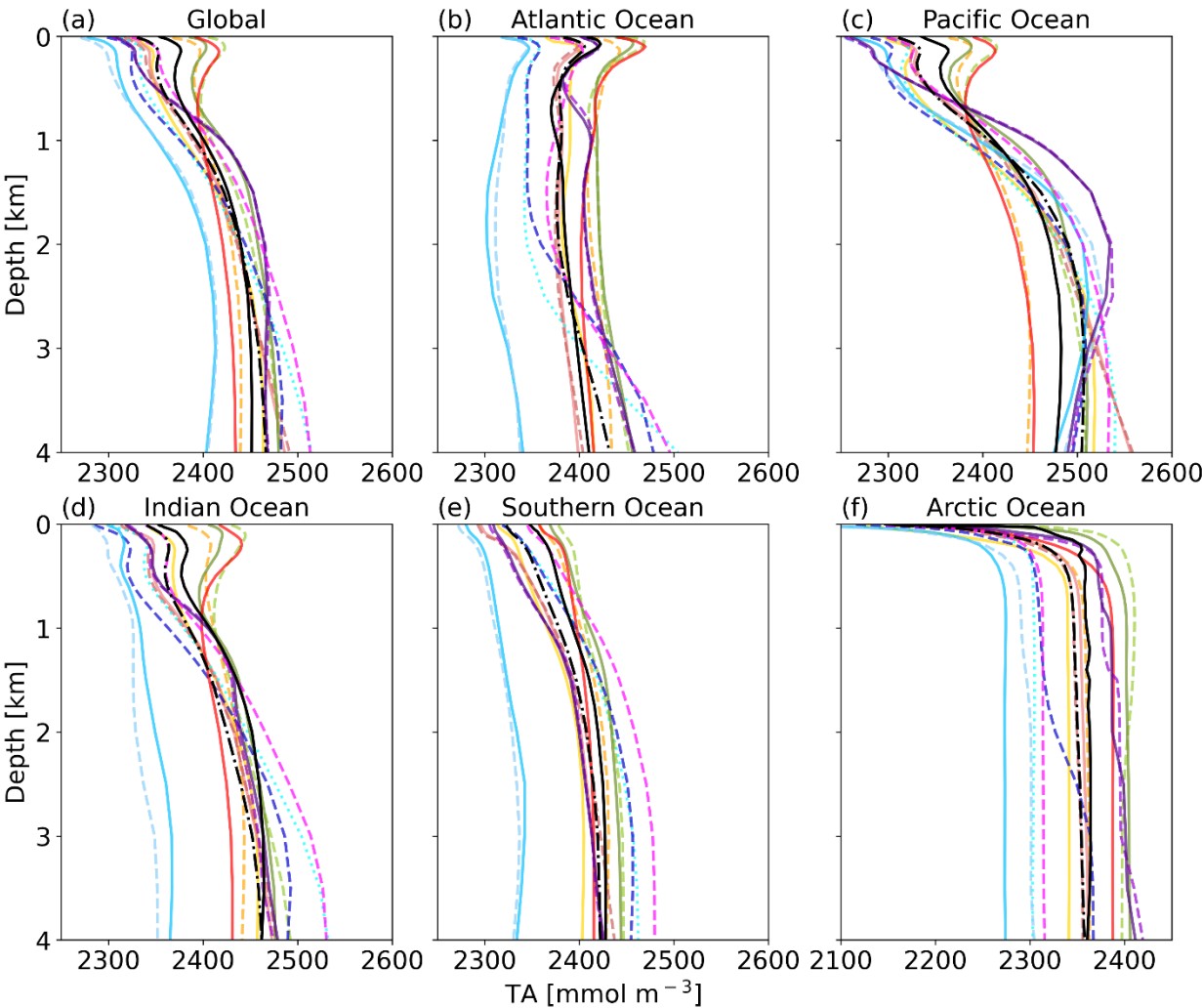

Figure 4: Global mean TA profiles for the major ocean basins. Color assignment is the same as in Figure 3.

The surface DIC patterns compared to GLODAP show very similar patterns to those for TA, both in general direction and local distribution (Figure 5). The global mean surface biases in TA compared to GLODAP range from -85 mmol m$^{-3}$ (-3.6%) to +50 mmol m$^{-3}$ (+2.1%), where the MMM bias is -25 mmol m$^{-3}$ (-1.1%) and for the global mean surface DIC the biases range from -55 mmol m$^{-3}$ (-2.6%) to 53 mmol m$^{-3}$ (+2.5%), with the MMM bias being -13 mmol m$^{-3}$ (-0.6%). TA biases likely lead DIC biases, as DIC can adjust through gas-exchange of $CO_2$ to maintain a surface chemical equilibrium with the atmospheric $CO_2$ concentration. Models with higher TA have higher DIC values and vice versa. We next investigate the origin of the models' alkalinity biases.

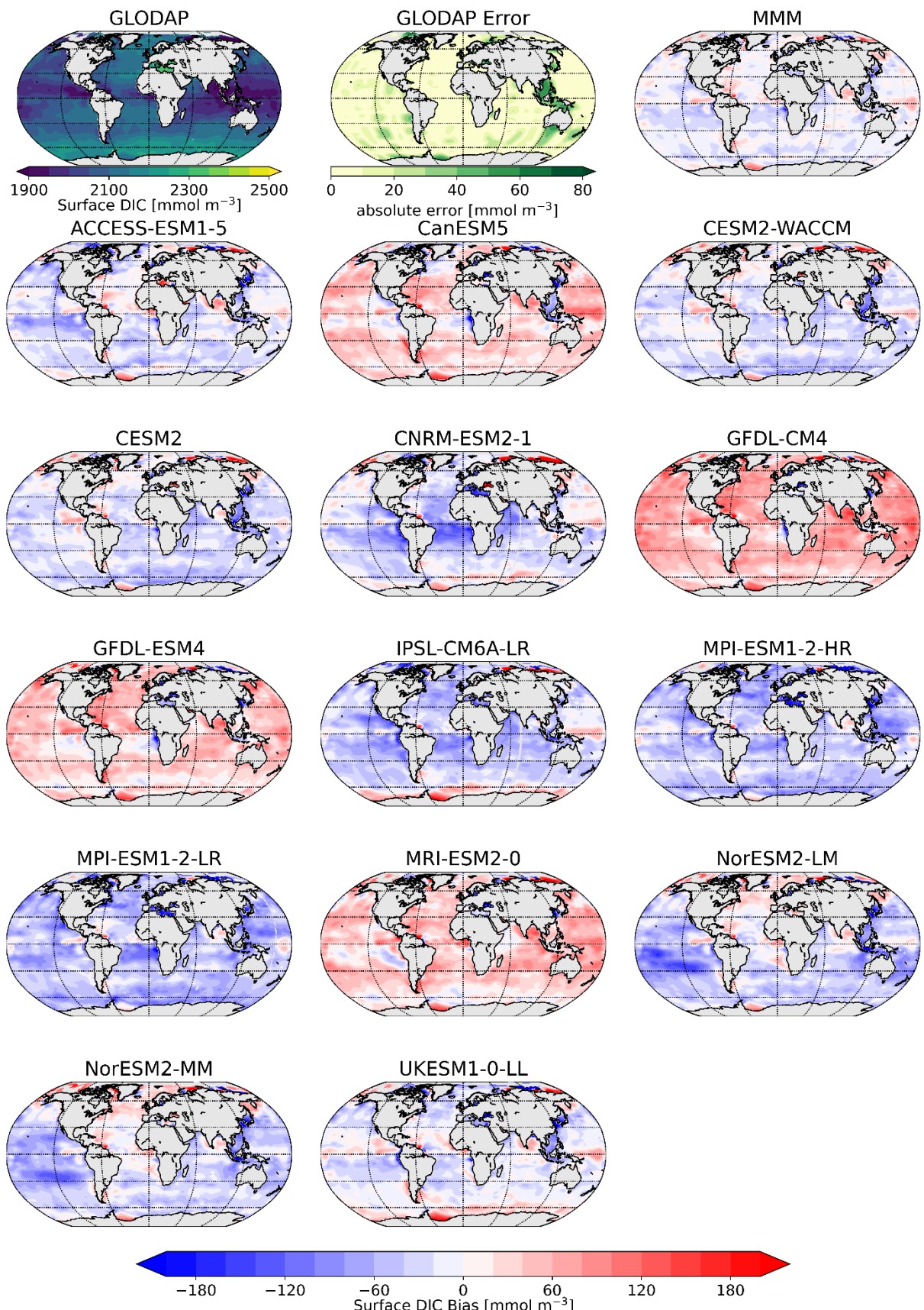

**Figure 5: Surface distribution of DIC in GLODAP (top left), its error estimate (top center) and the CMIP6 multi-model-mean (MMM) bias (top right), as well as the respective biases of the ESMs.**

## 3.2. Decomposition of the vertical alkalinity biases

The goal of the 'TA* method' (Koeve et al., 2014) is to separate the TA bias into contributions from 1) an inadequate representation of ocean physics or forcings (e.g., circulation, freshwater flow, evaporation, and precipitation), 2) the parametrization of calcium carbonate ($CaCO_3$) formation and dissolution and 3) the parametrization of organic matter remineralization processes. The first part, preformed alkalinity, includes the advection and upwelling of already biased water masses.

The decomposition of the TA biases (Figure 6a) shows that in the upper 1 km most of the models' alkalinity biases are due to their preformed TA (Figure 6b). Per definition, models with a negative surface TA bias have a negative bias in preformed TA. Below about 1,000 m depth $TA^0$ stays constant with depth. TA biases from the representation of organic matter remineralization processes are in the order of 5 to 10 mmol m$^{-3}$ and play only a negligible role in absolute terms (Figure 6c). The bias in TA from calcium carbonate dissolution in the interior ocean (Figure 6d) can be in absolute terms comparable to or even larger than the bias in preformed TA. The MRI model and the GFDL models have a small negative bias in TA* in the order of ~10-20 mml m$^{-3}$, relatively constant with depth. The MPI and NorESM models have a slight positive TA* bias in about the same order of magnitude, also relatively constant with depth, while the UKESM, the CNRM-ESM and the IPSL ESM exhibit TA* biases that increase with depth. The CNRM model has the largest TA* bias with about 100 mmol m$^{-3}$ at 4,000 m depth. CNRM-ESM2-1 and IPSL-CM6A-LR, have in common that they contain the same ocean model (NEMO) and the same biogeochemical model (PISCESv2). Dissolution in PISCESv2 is treated explicitly and is dependent on the calcite saturation state and the sinking speed for PIC is depth-dependent, while for other models the sinking speed is constant. In two of the models (of Figure 6d), MRI-ESM2-0 and UKESM1-0-LL, $CaCO_3$ is dissolved without a sediment, while the other models do have explicit sediment treatments where $CaCO_3$ is buried or dissolved, either depend on the calcite saturation state or a set rate (Planchat et al. 2023). A direct link of the treatment of $CaCO_3$ at the bottom to the bias at depths is not obvious in this case.

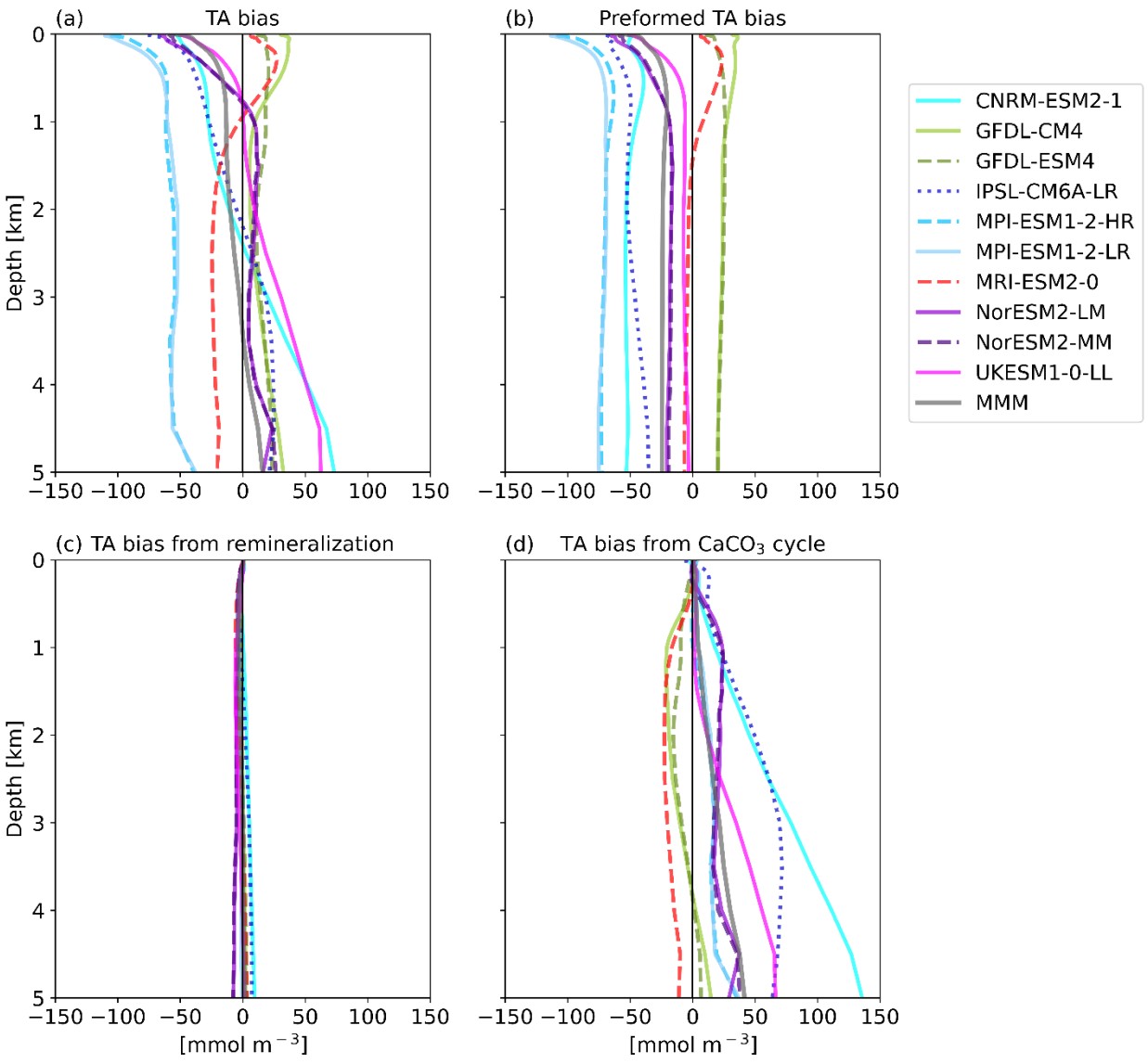

**Figure 6: Globally averaged depth profiles of biases in (a) TA, (b) preformed TA (TA⁰), (c) TA from remineralization (TAʳ) and (d) from calcium carbonate formation and dissolution (TA\*) in 10 CMIP6 models compared to the GLODAP climatology.**

### 3.3. Impact of biases on OAE efficiency

Biases in simulated surface TA and DIC have implications for the individual models' efficiency of OAE. By causing biases in the Revelle factor, they also result in biases in initial surface ocean $pCO_2$ reduction after alkalinity addition, and final $pCO_2$ values after equilibration with the atmosphere might differ. In order to evaluate the range of this sensitivity, a back-of-the-envelope-calculation was conducted, using the ESMs' surface TA, DIC, $pCO_2$ and an alkalinity addition of 100 µmol kg⁻¹ to calculate the full carbonate system before and immediately after alkalinity enhancement and after assumed air-sea equilibration (see methods). The results from this calculation (Figures

7b,d,e,f,g,h,i) together with the initial, unperturbed TA and TA-DIC-ratios (Figures 7a,c) are assessed for the ESMs and the MMM against the respective values for GLODAP.

The global mean Revelle Factor from the CO2SYS computation for the GLODAP dataset is with 10.19 the third lowest in our compilation and thus almost all ESMs have a higher Revelle Factor than the GLODAP data, ranging from 10.18 to 10.54 (Figure 7b). The Revelle factor is anti-correlated to the average TA-DIC-ratio (R=-0.99, Figure 7c). Also, the order of surface pH (R=-0.96, Figure 7g) and $pCO_2$ (R=0.97, Figure 7d) values corresponds largely to each model's rank in Revelle Factor (and thus also with TA-DIC-ratio). Models with a higher Revelle factor than GLODAP have a lower buffer capacity, which leads to already higher $pCO_2$ values (290 to 314 µatm) and lower pH (8.12 to 8.17) than in GLODAP ($pCO_2$: 292 µatm, pH: 8.16). Those models also show a greater initial reduction in surface ocean $pCO_2$ for the hypothetical addition of 100 µmol kg$^{-1}$ of TA (R=-0.99, Figure 7e) than GLODAP (-92 µatm), ranging from a 91 µatm to a 104 µatm decrease in pCO2. Models with a higher Revelle factor also have a higher uptake efficiency, $\eta CO_2$, (R=0.98, Figure 7f). The initial change in pH after alkalinity addition (Figure 7h) is about an order of magnitude larger than the change in pH after complete air-sea equilibration at constant atmospheric $CO_2$ (Figure 7i). The respective changes in pH (unequilibrated / equilibrated at constant atm $CO_2$) have a higher correlation to TA (R=-0.92, R=-0.99) than to the Revelle factor (R=0.83, R=0.63).

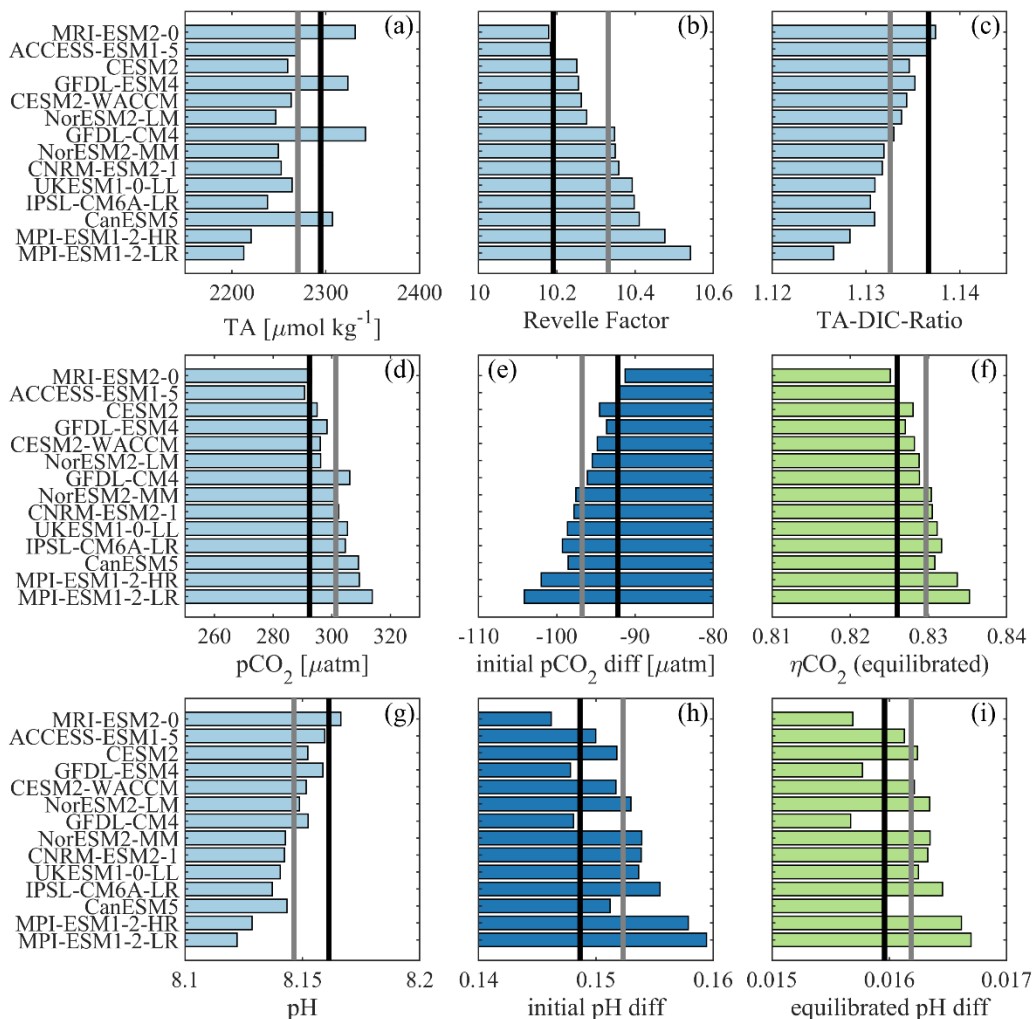

**Figure 7: Carbonate system parameters were computed for all CMIP6 ESMs, the MMM (grey line) and the GLODAP data (black line) with the CO2SYS toolbox. The results are sorted by Revelle Factor in ascending order for all panels. Shown are the TA (a), the Revelle factor (b), the TA-DIC ratio (c), initial pCO₂ (d), the difference in pCO₂ after a 100 µmol kg⁻¹ addition of TA (e), the uptake efficiency ηCO₂ (f), the initial pH (g), the difference in pH for constant DIC (h), and the difference in pH for constant pCO₂ (i). Light blue colors indicate the unperturbed mean state in the ESMs and GLODAP, dark blue colors the initial state after OAE and green colors the state after OAE and subsequent air-sea equilibration.**

In relative terms, we find that the ESMs' TA biases range from -3.6% to +2.1% with a mean of -1.1% and their DIC biases ranges from -2.6% to +2.5% with a mean value of -0.6% (Figure 1). Furthermore, the ESMs estimates of the initial pCO₂ decrease after a hypothetical TA enhancement by 100 µmol kg⁻¹ t ranges from -1.0% up to 13.0% (mean 5.1%) relative to GLODAP (Table S2). The controlling factor for this bias in initial pCO₂ reduction is in most cases the Revelle factor rather than the TA bias alone because the TA bias is always accompanied by a (partly) compensating DIC bias.

This simplified OAE example shows that for 12 out of 14 ESMs an increase of 100 µmol kg$^{-1}$ in TA would lead to a higher initial decrease in pCO$_2$ than observational data from GLODAP suggest. A higher sensitivity to TA changes due to a higher Revelle factor has also been shown in Hauck et al. (2016) during a decadal scale OAE simulation. We additionally calculated the effect of the additions of 200, 500 and 1,000 µmol kg$^{-1}$ of TA. The degree of pCO$_2$ difference overestimation decreases with the amount of TA added, but for a theoretical addition of 1,000 µmol kg$^{-1}$ of TA the maximum initial pCO$_2$ reduction overestimate with respect to GLODAP is still 8% (Table S2). We conclude that almost all ESMs might overestimate the initial pCO$_2$ difference in simulated OAE experiments. On the other hand, the CO$_2$ uptake efficiency computed with constant pCO$_2$ (equilibrated DIC) only differs by -0.1% to 1.1% (mean: 0.4%) from the GLODAP value, and the ESMs may thus represent equilibrium CO$_2$ uptake rather robustly.

The initial increase in pH after alkalinity addition is relatively large (Figure 7h, >0.1 pH units, i.e., on the same order of magnitude as the pH decrease since industrialization). However, after equilibration with the atmosphere (at presumed constant atmospheric CO$_2$), the lasting pH change is small (about 0.016, Figure 7i). These pH changes are in line with previous quantifications (e.g., Köhler et al., 2013, Hartmann et al., 2013, Hauck et al., 2016, rather independent of amount alkalinity added) and their small magnitude is the direct result of the additional carbon uptake from the atmosphere. In emission-driven simulations, where atmospheric CO$_2$ is substantially reduced through large applications of alkalinization, pH increases more substantially (e.g., by < 0.1 for an atm. CO$_2$ reduction of < 100 ppm, Lenton et al., 2018; by >0.3 for an atm. CO$_2$ reduction of >1000 ppm in a multimillennial simulation, Köhler, 2020). These findings call into question the common made statement that ocean alkalinization is unique as it 'simultaneously mitigates atmospheric concentrations of CO$_2$ and ocean acidification' (Burt et al., 2021; Ilyina et al., 2013; National Academy of Science, 2022). While ocean alkalinity enhancement allows for additional CO$_2$ uptake at a pH level that does not drop any further, a restoration/rise in pH is only possible if (a) the water mass is not in contact with the atmosphere (maybe for deep ocean applications) or (b) ocean alkalinization is efficient in reducing atmospheric CO$_2$, which is the driver of ocean acidification. The latter case, however, applies to all land- and ocean-based CDR methods that are efficient in reducing atmospheric CO$_2$, and thus ocean alkalinity enhancement is not unique in this regard.

## 4. Discussion and conclusions

We evaluated CMIP6 models regarding their large-scale biases in TA and DIC compared to the gridded data set GLODAP. Ten out of 14 ESMs underestimate surface TA (MMM: -25 mmol m$^{-3}$; i.e., -1.1%) and DIC (MMM: -13 mmol m$^{-3}$; i.e., -0.6%) with respect to observations. The range of the bias in TA is -85 mmol m$^{-3}$ (-3.6%) to 50 mmol m$^{-3}$ (+2.1%) and in DIC is -55 mmol m$^{-3}$ (-2.6%) to 53 mmol m$^{-3}$ (+2.5%). This is a reversal from the TA and DIC representation in CMIP5, where most models and the MMM overestimated these variables, and the absolute and relative errors were at least twice as large as in CMIP6 (Planchat et al., 2023). The direction of the bias and the relative biases of TA and DIC have a direct impact on the Revelle factor and the initial pCO$_2$ reduction of the surface ocean after alkalinity addition (and thus affect CO$_2$ uptake) and should be known when assessing model experiments simulating OAE or other NETs that directly affect the ocean's carbonate chemistry. Terhaar et al. (2022) also found that CMIP6 models overestimate the Revelle factor and propose that CMIP6 models underestimate the anthropogenic

ocean carbon sink from 1994 to 2007 by 9%, of which around 3% can be explained by the overestimation of the Revelle factor and the remaining 6% are related to the models' underestimation of the formation of mode and intermediate water in the Southern Ocean (Terhaar et al., 2021).

It is helpful to understand the contributions of the physical and biological - the soft tissue and calcium carbonate - pumps to these TA biases in ESMs. The value of decomposing the carbon pump has already been recognized in previous studies (e.g., Sarmiento and Gruber 2006; Kwon et al. 2009); however, there is not a common standard to achieve this decomposition. Here, we separated the global mean vertical TA bias into contributions from preformed alkalinity ($TA^0$, physical pump), remineralization ($Ta^r$, soft tissue pump) and alkalinity from calcification and

carbonate dissolution (TA*, $CaCO_3$ pump) following Koeve et al. (2014). This decomposition method aims to compute the physical contribution to the alkalinity distribution explicitly, similar to Oka et al. (2020) and contrary to Sarmiento and Gruber (2006) and Planchat et al. (2023). An advantage of this method is that the preformed alkalinity is computed for each grid point and therefore is resolved spatially. In their presentation of the method, Koeve et al. (2014) note that the computation of TA* according to equations (2) to (4) reproduces tracer-based simulated TA*

robustly in most of the global ocean, but that higher uncertainties occur in the Atlantic and in the 500 m to a 1,000 m layer in the Pacific and Indian ocean. Here, we only focused on the TA* results for the global mean ocean. A caveat that was mentioned by Koeve et al. is that AOU is known to overestimate true oxygen utilization by 20–25 %. Hence our $TA^r$ computed from AOU probably also overestimates by this percentage. But $TA^r$ is rather small and here we focus on the implication of the TA* biases in ESMs and potential remedies for these biases.

The result from our TA* analysis is that especially in the upper ocean the global distribution of TA in ESMs is largely determined by preformed TA which is set by ocean model physics (advection, overturning, mixing, etc.), although this performed TA is not purely physical but also contains the physical redistribution of already biased TA. In the sub-surface and deep ocean, biases in TA are also driven by the $CaCO_3$ cycle, while contributions from remineralization are negligible. Although Planchat et al. (2023) do not assess alkalinity biases due to the physical carbon pump, they

also point to a larger contribution of the carbonate pump relative to the soft tissue pump (remineralization) to the (normalized) TA biases. The model processes involving the physical distribution of TA are tuned to achieve the best overall model performance and it could be tested whether a tuning to improve TA would support this goal. The findings regarding the contribution to the TA biases from the $CaCO_3$ cycle simulation suggest that improving the parametrizations of biogeochemical processes that are sources and sinks of TA, e.g., calcification, remineralization of

sinking detritus, chemical dissolution of calcium carbonate, biological $CaCO_3$ formation and dissolution, etc. would be beneficial. Since the bias in TA from remineralization is small in all models, parametrizations that affect the carbonate chemistry are the most practical lever to improve the TA distribution for most models. This, in turn, needs a much-improved process understanding of $CaCO_3$ dissolution in microenvironments such as aggregates, zooplankton and fish guts above the calcite and aragonite saturation horizons (Sulpis et al., 2021; Jansen and Wolf-Gladrow, 2001;

Salter et al., 2017) from field and laboratory studies in order to mechanistically represent these processes and how they might be altered in a high-$CO_2$ ocean. In the absence of this mechanistic understanding, some suggestions to reduce TA biases are:

Possibilities for model tuning:

- If TA is low at the surface, decreasing the calcification (rate) within realistic limits or increasing near-surface dissolution could be beneficial (Gangstø et al., 2008; Gehlen et al., 2007).
- If the calcite dissolution is prescribed to increase with depth (Yamanaka and Tajika, 1996) this process could be tuned to better match the observed vertical distributions of calcite or TA.

Possible expansion of model parametrizations:

- If calcite dissolution is formulated as (mostly) saturation-dependent and is therefore (close to) zero above the calcite saturation horizon, a term should be implemented that encompasses dissolution processes that have been observed to occur above said horizon, e.g., calcite dissolution in microenvironments like marine snow and zooplankton guts (Sulpis et al., 2021). It was shown that the acidic environment in guts of starving copepods can dissolve up to 38% of the calcite taken up by grazing (White et al., 2018). For non-starving copepods this value was somewhat lower (Pond et al., 1995; Jansen and Wolf-Gladrow, 2001).
- In addition to those processes, it is known that aragonite and high-magnesium calcite have a shallower saturation horizon than calcite and contribute to upper-ocean calcium carbonate dissolution (Sabine et al., 2002; Gangstø et al., 2008; Barrett et al., 2014; Battaglia et al., 2016). Almost all models only simulate calcite explicitly (Planchat et al. 2023) which is a deficit since Buitenhus et al. (2019) proposed that aragonite producing pteropods might contribute at least 33% to export of $CaCO_3$ at 100 m and up to 89% to the pelagic calcification. Although exact numbers might be subject to reevaluation when more data becomes available, a carbon cycle formulation expanded to also simulate aragonite (formation and dissolution) may be beneficial for a more realistic alkalinity distribution.
- The representation of $CaCO_3$ treatment at the bottom-sediment interface (dissolution, sedimentation, sediment weathering) is important for the total alkalinity budget and also for upper ocean alkalinity especially in more shallow regions where alkalinity-enriched waters (through dissolution) can recirculate to the upper ocean more quickly (Gehlen et al., 2008).

The back-of-the-envelope calculations of the ESMs' carbonate system states revealed that all but two of the models have a higher global mean Revelle Factor than calculated from GLODAP, correlated with a higher TA-DIC-ratio than suggested by observations (see also Terhaar et al., 2022). For a hypothetical addition of 100 µmol kg$^{-1}$ TA this bias leads to an overestimation of the initial pCO2 reduction by up to 13% (affecting $CO_2$ uptake from the atmosphere). The addition of just 100 µmol kg$^{-1}$ TA is actually at the very low end of the spectrum used in past and current OAE experiments in models and in mesocosms (Hartmann et al., 2022; Ferderer et al., 2022). This calculation is a simplified exercise since gas exchange between ocean and atmosphere is not accounted for nor the potential precipitation and sinking of calcium carbonate (Hartmann et al., 2022). The $CO_2$ uptake efficiency factor, $\eta CO_2$, relates changes in surface DIC to alkalinity input. We computed this metric here with constant $pCO_2$ after alkalinity addition which suggests complete equilibration and neglects any reduction in atmospheric $CO_2$ due to OAE. Studies suggest that the time scale and efficiency of the equilibration can differ immensely depending on the ocean region. He and Tyka (2023)

found that after one year $\eta CO_2$ varied between 0.2 and 0.85 and that after 10 years most locations showed an uptake fraction of 0.65–0.80. Jones et al. (2014) quantified the mean global air-sea equilibration timescale for $CO_2$ at 4.4 months (range 0.5 to 24 months regionally). Bach et al. (2023) suggest a pragmatic time scale of 10 years for a 95% DIC equilibration after OAE measures. It is within this range of suggested equilibration time scales that the differences in simulated pCO2 change between ESMs are important.

The results of our idealized calculation also highlight the need to monitor at least two carbonate system variables to characterize the full carbonate system after alkalinity addition in a potential real world application. Knowing the amount of alkalinity added and then monitoring $pCO_2$ with an autonomous sensor will not be sufficient to characterize the full carbonate system and the level of equilibrium reached, particularly as alkalinity and carbon will be subject to transport through mixing and advection. Autonomous sensors with high accuracy are currently only available for $pCO_2$, whereas alkalinity sensors are not commercially available (see review in Ho et al., 2023) and pH sensors do not have high enough accuracy (Wimart-Rousseau et al., 2023). This poses a challenge for monitoring, reporting and verification (MRV) that may be tackled through (i) measuring discrete water samples until technical advances make autonomous measurements of two carbonate system variables possible or (ii) using models of high fidelity. In order to fully capture the effect of OAE on atmospheric $CO_2$ concentration and the model spread related to biases stemming from circulation and biogeochemical assumptions, model OAE experiments need to be performed in a suite of fully coupled emission-driven ESMs with a precise protocol and with realistic representation of the carbonate pump, including $CaCO_3$ dissolution above the carbonate saturation horizon, which is not even sufficiently understood in the real world (Sulpis et al., 2021).

**Acknowledgements**

This project has received funding from the European Union's Horizon 2020 research and innovation program under grant agreement number 869357 and from the Initiative and Networking Fund of the Helmholtz Association (Helmholtz Young Investigator Group Marine Carbon and Ecosystem Feedbacks in the Earth System [MarESys], grant number VH-NG-1301) . Furthermore, this work used resources of the Deutsches Klimarechenzentrum (DKRZ) granted by its Scientific Steering Committee (WLA) under project ID ba1103.

**Code availability:**

The CO2SYS matlab toolbox is available at
https://cdiac.ess-dive.lbl.gov/ftp/co2sys/CO2SYS_calc_MATLAB_v1.1/ .

**Data availability:**

All CMIP6 model output has been downloaded from the data sources given in Table 1.

**Author contributions:**

JH is PI, CV and PK are co-PIs of this sub-project contributing to the EU project OceanNETs (money aquisition). CH performed the data analysis, preparation of the figures and led the writing of the draft. All co-authors

contributed to draft writing by editing the initial version.

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
