# Peer review of "Alkalinity biases in CMIP6 Earth System Models and implications for simulated CO2 drawdown via artificial alkalinity enhancement"

_Biogeosciences, 2023_

## Referee Comment (RC2)

**Review of Hinrichs et al. (2023)**

**Overall comment**

In overall, this work is interesting and innovative in its approach to bias assessment in OAE modelling studies using ESMs. Indeed, the study pushes a CMIP6 bias analysis (essentially focused on alkalinity, but also on DIC) towards a concrete, but idealized, CDR experiment. The approach used to decompose the alkalinity bias is broadly consistent with Planchat et al. (2023), thus reinforcing the message to the modelling community around the representation of alkalinity and the carbonate pump in ESMS. The writing of the manuscript could nevertheless be partly improved, in particular some parts could be restructured to gain clarity. Many technical corrections should be made.

**Specific comments**

    *A. Main comments*

      1.  In the decomposition of the alkalinity bias, it is repeatedly mentioned that "preformed alkalinity" refers to physical biases for the ocean, but it also contains biases of both organic matter and $CaCO_3$ production (e.g., l. 17 ; l. 121-125 ; l. 218 ; l. 291 ; l. 294)

      2.  The direct link made with the $CO_2$ uptake in the OAE idealized experiment is confusing for me (e.g., 274, l. 327): a relative $pCO_2$ difference should not directly be a relative $CO_2$ uptake difference, or shoud it?
         According to me, we have:

$$\frac{dDIC}{DIC} = \frac{1}{Re} \cdot \frac{dpCO_2}{pCO_2}$$

         An ESM with a greater relative variation of $pCO_2$ resulting from alkalinity addition, also has a greater Revelle factor. So, the product of the right member of the above equation should at least partly compensate. To access a first approximation of the $CO_2$ uptake associated with alkalinity addition, the relative variation of DIC should be expressed in function of the relative variation of alkalinity. I think that with what is currently offered, it is possible to assess the bias resulting from OAE in ESMs concerning acidification, but I'm confused by the direct link made with the $CO_2$ uptake.

    *B. Minor comments*

The second part of the abstract could be improved and made more readable. In particular, I do not understand: "We find that the degree of compensation of DIC and alkalinity biases at the surface is more important for the marine $CO_2$ uptake capacity than the alkalinity biases themselves." (l. 25-26); and l. 27-29 are too heavy, perhaps misplaced, and with a unit error (mmol/kg instead of mmol/m3 as l. 209-211).

For the sake of clarity and consistency throughout the manuscript, I would suggest that alkalinity and DIC should always be shared in mmol/m3 rather than mixing mmol/m3 and umol/kg. It would simply be necessary to clarify how the conversions are done for GLODAPv2 and for using CO2SYS in the methodology (see l. 28-29, l. 149-152, Fig. 1 etc.).

Table 1: It would be great also to share the ensemble member for each ESM, rather than writing "the first available ensemble member" l. 109, which is slightly confusing.
The marine biogeochemical model of CNRM-ESM2-1 (not CNRM-ESM-2-1) is PISCESv2-gas. The one for IPSL-CM6A-LR is PISCESv2. The one for NorESM2-LM/MM is iHAMOCC.
Why do you precise the grid only for a few ocean models? If you wish to precise it, add it in the "ocean horizontal resolution" column and just precise maybe whether it is a tripolar grid or not.

l. 109-119: These lines would benefit from restructuring.

The TA* method: are there limitations to this decomposition of the biases? What are the advantages of it rather than the one suggested in Sarmiento and Gruber (2006)? Not using DIC?
Eq. (3) and (4): why do you use phosphate for PO in Eq. (3) and then nitrate in Eq. (4) for $TA^r$ rather phosphate (or nitrate) in both cases?
l. 139, you explicitly mention the fact that TA* "is computed as residual after rearranging Eq. (2)". This is questioning, since it is finally on the dissolution of $CaCO_3$ that we have the more certainty over its affects on alkalinity (+2 eq for 1 eq of $CaCO_3$ dissolved). On the contrary, the effect of remineralisation, for example, is more complicated to extract from ESMs because of the N-reactions and whether or not they are taken into account in the biogeochemical models. It is therefore perhaps a pity not to take advantage of the ease with which the carbonate pump can be used in terms of its effect on alkalinity with this method.

l. 156-157: Could you explain why you have chosen these values rather than the mean surface ESM values for instance?

l. 166-168: This sentence should be part of the methodology.

l. 190 and Fig. 3b: The very high $TA_n$ value observed at the ocean surface for GFDL-ESM4 is linked to very low salinity values (even possibly null) in the Baltic Sea if I remember well. You might be able to neglect this closed sea for this ESM when averaging to avoid this issue. You could also use a salinity threshold maybe.

Fig. 3 and 4: I would combine both figures so that the first panel is not repeated.

l. 201: It could be mentioned that the bias observed for CNRM, IPSL and UK is associated with the ocean model NEMO.

l. 211: "TA biases likely lead DIC biases, as DIC can adjust through gas-exchange of $CO_2$". I suggest to add at the end "to maintain a surface chemical equilibrium with the atmospheric $CO_2$ concentration". Besides, this aspect could be slightly more highlighted to understand why you decompose the alkalinity biases and not the DIC biases in addition, whereas you consider both surface alkalinity and DIC in the following with CO2SYS.

l. 221-230: This paragraph should be rewritten.
It should be readable and meaningful independently of Fig. 6 (avoiding "are shown in Figure 6" for instance).
The first sentences are too repetitive compared to what was already described in the previous sections.
l. 226-228: This sentence is out of topic as you are talking about biases in this section.
It would be great also to share some values (in absolute and/or %).
Fig. 6: The addition of MMM would be meaningful on these plots.

l. 237: I am not convinced of the veracity of the causal sequence (cf. "thus"). Could you detail it?

l. 240-242: "All panels are sorted by Revelle factor in ascending order." This is enough in the legend.
"The Revelle factor … at the ocean surface." should be part of the Introduction.

Fig. 7: Could you set "MMM" in bold so that we can easily spot it?
I also suggest to keep the x-axis increasing in the last panel, to avoid confusions.

l. 260-261: Too repetitive with what was already said for Fig. 1.

l. 262: It is less than 13.0 % in Fig. 8, I think: t should be about 10.7 %.

l. 263: It would be great to have also the TA bias in Fig. 7 with the ESMs ordered in the same way.

Fig. 8: Could you set "MMM" in bold so that we can easily spot it?

Could you adapt the y-axis scale to avoid the blank space to the top?

Discussion and conclusions: In overall, this section could be improved in terms of organization and content. There are some repetitions and it is sometimes difficult to follow you.
In particular, l. 310-323 are quite messy for instance and the points overlap each other sometimes. In the first point, are you pointing towards including aragonite as well as calcite? What would be the effect of reducing the calcification rate in the Southern Ocean, where deep waters are upwelled? l. 320: are you talking about calcite dissolution (effectively sometimes explicitly modelled) or calcite production (always implicitly modelled in that case)? Another important point, which is not mentioned, is the burial and dissolution at the seafloor.

Fig. S2: It is not compared to the GLODAP climatology here, as opposed to Fig. 6.

Table S1: Not useful, I think.

There might be some issues in the References. There is at least one for Planchat et al., 2023, since some authors are missing.

**Technical corrections**

*The following remarks are valid for the whole manuscript, and only a few occurrences will be mentioned below:*
- There is a regular lack of punctuation, mainly commas.
- Beware of citations: a number of references appear with double brackets in the text.
- In intercomparison studies of ESMs, it may be more accurate and preferable to refer to ESMs throughout the study rather than models.
- If you decide not to put a space before '%', do so everywhere.
- I suggest to write 'TA* method' instead of 'TA* Method'
- The p in $p$CO$_2$ should be in italic
- When the abbreviation/acronym has been defined, it can then be used directly (especially in the legend of the figures)
- Try to keep the text independent of the figures and mention the figures in parentheses instead of within the text directly
- Write "Revelle factor" instead of "Revelle Factor"
- Always write "TA-to-DIC-ratio" instead of mixing the way you mention it

l. 4: 1 should in exponent

l. 9-15: commas are missing

l. 10: … on alkalinity …

l. 10: That is why, in the search for …

l. 11: … Ocean Alkalinity Enhancement (OAE) …

l. 12: … exists on how …

l. 13: … 14 CMIP6 Earth system models (ESMs) …

l. 17: … shows …

l. 36: … overestimate it in the …

l. 40-42: commas are missing

l. 45: … species composing the so-called …

l. 45-47: Suggestion: The oceanic uptake of anthropogenic carbon leads to an increase in aqueous $CO_2$ and thus DIC. By changing the chemical equilibria between the carbonate species, this results in ocean acidification with a decrease in pH.

l. 48: … over acids (proton donors) …

l. 48: … role in the partitioning of the DIC pool, especially in the form …

l. 49-56: Very nice paragraph, but the first sentence could potentially be improved. Suggestion: The carbonate system is key in driving the seawater ability to resist a change in its chemistry, also called its buffering capacity. In particular, the Revelle factor is the sensitivity of $pCO_2$ to changes in DIC. A low Revelle factor indicating a high buffering capacity and vice versa.

l. 55: … the potential $CO_2$ uptake …

l. 68: … much research has focused on …

l. 72: … (OAE; Köhler et al., 2013 …

l. 81: … set-ups (e.g., Ilyina et al., 2013; … ; Burt et al., 2021).

l. 86-87: … underway, or in planning, seeking to apply … for OAE (e.g., Butenschön et al., 2021), highlighting the importance and urgency of a robust ESM evaluation.

l. 93: They report an improvement in …

l. 109: "the first available ensemble member" is confusing and not precise.

l.111: (CDO; Schulzweida, 2022)

Eq. (1): Precise that S is the salinity

l. 115-119: Suggestion: … against gridded observational products: (i) TA, DIC and pH from the GLODAPv2.2016b Mapped Climatology (GLODAP in the following; Lauvset et al., 2016); (ii) oxygen and nutrients from the World Ocean Atlas 2018 dataset (WOA; Garcia H.E., 2019) and GLODAP; and (iii) salinity and temperature from the Polar science center Hydrographic Climatology (PHC3.0; Steele et al., 2001) and WOA. For the evaluation of global mean vertical profiles, the model data are interpolated onto the same 33 vertical levels used in the GLODAP climatology.

l. 125: … calcium carbonate ($CaCO_3$) dissolution (TA*)

l. 128 : potential temperature

l. 129 : if NO is mentioned, then it should be defined to explicitly mention the difference with PO.

l. 131, 136: could you add references for the ratios $r_{(…)}$ that you consider?

l. 131-132: "onto upper ocean TA values" and then "interior ocean" is a bit contradictory

l. 140: … to 10 of the 14 CMIP6 ESMs (… NorESM2-MM and UKESM1-0-LL)

l. 143: 2.3. Theoretical model sensitivity to OAE

l. 145: OAE. Thus, …

l. 147-148: Suggestion: this toolbox, from the combination of two of the carbonate system variables, compute the entire ocean $CO_2$ system.

l. 149-152: Suggestion: … 1,026 kg/m3 (Fig. 1). First, we evaluate the CO2SYS output fields : Revelle factor, pH and $p$CO$_2$ (partial pressure of CO$_2$ in seawater) based on the CMIP6 outputs. Then, we assess the changes…

l. 155 … (MMM). Black vertical … value. (b) Same …

l. 156: potential temperature

l. 164: … at the surface (Fig. 1a and 2)

l. 165: (Fig. 1a and 2)

l. 170: … in NorESM-LM/MM and CanESM5; Fig. S1).

Fig. 2: Panel 1, title: GLODAP
Panel 2, title: GLODAP error; colorbar title: absolute error.
General colorbar title: Surface TA bias [mmol/m3]
Legend: Surface distribution of TA in GLODAP (top left), its error estimate (top center) and the CMIP6 multi-model-mean (MMM) bias (top right), as well as the respective biases of the ESMs.
Same comments for Fig. S1, where TA$_n$ should be directly mentioned.
Same comments for Fig. 5.
Try to homogenize the legends too

l. 177: … the increase of TA with depth deeper …

l. 179: … TA overall. This indicates that their global inventory of TA is respectively too low and too high …

l. 182: … in the GFDL ESMs …

l. 185-186 1,024; 1,028; 1,035

l. 187: … at depth, or vice versa, …

l. 189: … in the upper ocean. The surface minima …

Fig. 3: Could the black and grey lines have no transparency and be to the forefront so that we can clearly see both of them?
Legend: … and TA$_n$, (b) …

l. 199: … by the ESMs, referring to circulation biases.

l. 200: … Ocean), between … depth, …

l. 208: … and local distribution …

l. 216: 3.2 Decomposition of the vertical alkalinity biases

l. 219: … (CaCO3) dissolution and …

Fig. 6: subplot titles: I suggest to simply write "TA/TA$^0$/TA$^r$/TA* bias"

l. 235: Impact of biases on OAE efficiency

l. 236: … surface TA and DIC …

l. 238: … was conducted, using surface TA and DIC, to calculate the full carbonate system (see Methods and Fig. 1).

l. 248: … GLODAP data, ranging from …

Fig. 7: … CO2SYS toolbox. The results …

l. 269: … OAE idealized experiment shows that 12 out of 14 ESMs …

l. 278-279: … mmol/m3; i.e., YY %)

l. 294-295: In the sub-surface and the deep ocean, biases in TA are also driven by the CaCO3 dissolution, while contributions from remineralization of organic matter are negligible.

---

## Author Comment (AC1)

Overall I found this to be an interesting, well-written and illuminating paper that I think will help spur improvements in model development. The separation of the TA biases into preformed TA, remineralization TA and CaCO3 TA is also very useful and points to concretely implementable improvements, especially in the treatment of CaCO3.

We thank the reviewer for their thoughtful und thorough review and helpful suggestions. We addressed the reviewer comments below in blue.

I recommend publication with some minor revisions (see below).

My two major comments are:

Line 254 & Figure 7

As the authors point out the biases in Revelle Factor are of great importance to mCDR. An additional metric of this that would be straightforward to add using CO2SYS and of great value to folks investigating the feasibility and cost of ocean alkalinity enhancement is the uptake efficiency factor (In our work we like to call this $\eta CO2 = \partial DIC/\partial Alk$ at constant pCO2, see https://doi.org/10.1039/D1EE01532J and https://bg.copernicus.org/articles/20/27/2023/). The metric simply indicates the number of moles of CO2 taken up per mol of Alkalinity added after full equilibration (for an infinitesimal increase in Alk) and is generally ~0.8 though it is quite dependent on location (see for example He et al., 2023, https://bg.copernicus.org/articles/20/27/2023/).

I think this number is very practical because it directly represents an efficiency loss going from some alkaline substance to actual CO2 drawdown and thus enters any cost estimates. Thus I would be very curious to see how model biases affect this metric, even if just expressed as a global average or surface average.

We calculated the instantaneous uptake efficiency $\mu CO_2$ also with CO2SYS and added the results to Figure 7:

[Figure]

*Figure 7 NEW: Carbonate system parameters were computed for all models, the MMM (grey line) and the GLODAP data (black line) with the CO2SYS toolbox, based on the two input parameters global mean alkalinity and DIC. The results are sorted by Revelle Factor in ascending order for all panels. Shown are the Revelle factor (a), the TA-DIC ratio (b) pH (c), $pCO_2$ (d), $\mu CO_2$ uptake efficiency (e), and difference in $pCO_2$ after a 100 $\mu mol\ kg_{-1}$ addition of TA (f).*

As one can see the uptake efficiency in this case hovers around ~0.62 for the models and GLODAP. Figure 5 in https://bg.copernicus.org/articles/20/27/2023/ suggests that it would

take some time to get the expected values of ~ 0.8. Still, the models exhibit some differences in this instantenous uptake efficiency that reflects their initial state in TA, DIC and pCO2.

In response to Reviewer 2, we added TA in the same order to Figure 7 (and removed pH for now), this shows more clearly how the initial state of TA directly influences the uptake efficiency after alkalinity addition, while the effect on the pCO₂ difference is further modified by the Revelle factor and the PCO₂ initial state :

[Figure]

L317ff and Figure 6, panel (d). There is clearly a large amount of difference in TA* between models and also in some models these biases are clearly depth-dependent while in others they are less so. This is one of the major insights of this paper. Not being familiar with the details of each of the models tested, I am very curious about whether there is any pattern or correlation between the sophistication of the CaCO3-cycle-model in each GCM and the amount and type of bias observed ?

E.g. The blue-ish models mostly overestimate TA at depth - do they have something in common in the way they treat the CaCO3 dissolution?

The two models that show the highest bias in the CaCO3 cycle in Figure 6d, CNRM-ESM2-1 and IPSL-CM6A-LR, have in common that their ocean model is NEMO and the biogeochemical model is PISCESv2. Dissolution in PISCESv2 is treated explicitly and is dependent on omega and the sinking speed for PIC is depth-dependent, while for other models it is constant. More details on the model equations for all CMIP6 models can be found in Planchat et al. (2023) (https://bg.copernicus.org/articles/20/1195/2023/bg-20-1195-2023.pdf).

Do any of these models treat the natural occurrence and distribution of CaCO3 sediments

explicitly (see work by Sulpis et al and others for maps of this) ? or do they only account for precipitation and redissolution ? If not, then perhaps there is a spatial correlation between TA biases and occurence of CaCO3 sediments? I'd love to see more discussion of this phenomenon - it's very interesting! The discussion on L294-330 is in very general terms rather than looking at algorithm differences between the specific models that could explain the differences.

In two of the models (of Figure 6d), MRI-ESM2-0 and UKESM1-0-LL, CaCO3 is dissolved without a sediment, while the other models do have explicit sediment treatments where CaCO3 is buried or dissolved, either depend on omega or a set rate (Planchat et al. 2023). A direct link to the bias at depths is not obvious in this case. Since our study is more focused on biases at the surface and the subsequent implications for ocean alkalinity enhancement, we did not go into more detail here.

Figure Style comments:

As I was parsing the figures I felt some improvements in the plots could make better use of the space, aid visual parsing and generally make the paper even easier to follow. Please take these as suggestions, perhaps try them out and see if you like them.

Figure 1: Maybe expressing the MMM as a (say, dashed) line rather than an additional row would be more intuitive and allow visual comparison of each model vs the MMM ?

Figure 1: does the thickness of the GLODAP line have meaning (e.g. a standard error) or is it incidental ? If a standard error for the GLODAP measurement is known or can be computed it would be neat to use the thickness (using a semi-tranparent color) this way (unless the certainty is so high that it reduces to a thin line of course). I think this is important as a large GLODAP uncertainty could change or weaken the conclusions.

Figure 1, Line 264 As you note Alk and DIC are highly correlated, and they are compensating variables in the carbonate system, with respect to pH, pCO2 etc.

An alternative for the two panels in Figure 1 would be to plot both together as a scatter graph with DIC on the x-axis and Alk on the y axis (or vice versa). This way the exact same information is displayed but the extent of the correlation is visually immediately apparent as well. The scatter points could be labelled directly on the graph with a floating text for example. Error bars on each pint could indicate the variance of these values over the surface average.

Thank you for the suggestion. In response, we replaced Figure 1 with a scatter plot of TA versus DIC, as we agree that this might be more intuitive. This figure also contains an error estimate for the global mean GLODAP data. While the MMM is almost within the estimated GLODAP range, we see that the difference for individual models can be quite large. We also

see that TA and DIC biases are highly correlated.

[Figure]

*Figure 1 NEW: Global mean surface total alkalinity (TA) [mmol m⁻³] of the 14 CMIP6 models, the multi-model-mean (MMM), and GLODAP including its error estimate versus dissolved inorganic carbon (DIC) [mmol m⁻³].*

Figure 2&5: "Absolut error" → "Absolute error"

This has been corrected.

Figure 2&5: If the whitespace between globes could be reduced at all, that would make everything bigger and easier to parse (It's already tricky without looking at the PDF on a large screen).

Space between the subplots has been reduced as much as possible.

Figure 3: Ah, I see here there is a GLODAP error estimate. Great! Could this be added to Figure 1 also ?

Yes, this has been added to Figure 1. Thank you for the suggestion.

Figure 3&4: Style considerations: For a colour-blind person (like myself) it is nigh impossible to know which line is which, among similar shades/hues. I would recommend blending color with different dash/dot patterns to help with this. Perhaps the error (currently dashed lines) can be given simply by a transparent shaded area ?

We added linestyles to the profiles and indicate the GLODAP error estimate with shading.

[Figure]

Also, a lot of features of this graph occur in the upper 0.5km and are visually cramped in a very small area. For the same reason that most models have non-uniform vertical z slices, perhaps it would be possible to plot the vertical axis on a log scale or a mixed log-linear scale. Or split the graph into two linear regions, one for 0-500m and one 500-4000m ?

We would like to keep this figure as is since we were interested in how well the profiles overall match.

Figure 4(f) I'd adjust the x axis to not clip the values at shallow depth. Ah - I see all the X axes are coordinated. Hmmm. Not sure how to solve this. How low do the TA values go in the Arctic Ocean (last panel)? Perhaps one could plot all these as Deltas from GLODAP the same way that Figure 6 ? That might help with the dynamic range of the x axis (which IMHO does not necessarily have to be the same for each subpanel). I thought Figure 6 was very nice.

[Figure]

We adjusted the y-axis for the Arctic in Figure 4f and applied the linestyles here as in Figure 3 as well as added a line for MMM.

Figure 7: Again, using the thickness of the GLODAP vertical line to indicate variance would be neat.

[Figure]

The estimated error for the GLODAP values based on the TA and DIC GLODAP errors are very small for the computed parameters. Here, we added dashed vertical lines next to the black GLODAP line. We think the figure would be more clear without the dashed lines though.

Figure 8: I found this figure rather difficult to parse. Because the three different variables have such different dynamic range on the %-scale, especially the second one (TA-DIC ratio) is virtually impossible to read off. Is this figure really necessary ? I feel like most of the information content is already contained in Figure 7.

We will drop Figure 8 and instead refer to the percentage values listed in Supplement Table S2.

L87: Some other recent studies that would be worth including that also aim to be more realistic than the earlier large-scale uniform OAE simulations, i.e. near-coastal or ship-track-based releases or regional assessments.

https://doi.org/10.1002/2017EF000659
Model-Based Assessment of the CO2 Sequestration Potential of Coastal Ocean Alkalinization, Feng, Koeve, Keller, Orschlies 2017

https://doi.org/10.1029/2022EF002816
Simulated Impact of Ocean Alkalinity Enhancement on Atmospheric CO2 Removal in the Bering Sea, Weng et al., 2022

https://doi.org/10.5194/bg-20-27-2023
Limits and CO2 equilibration of near-coast alkalinity enhancement, He and Tyka, 2023

https://doi.org/10.5194/egusphere-egu23-9305
Atmospheric CO2 removal by alkalinity enhancement in the North Sea, Liu et al. 2023

L87 has been appended to include most of the above suggested references and now reads:

Now, more and more projects are underway or in planning that seek to apply more realistic scenarios for OAE e.g., in regional OAE applications (Butenschön et al. (2021), Wang et al. (2023) or coastal applications (Feng et al. (2017), He and Tyka (2023)), which is why a model evaluation is even more important.

L149: Since the carbonate system isn't linear wrt TA and DIC, does it make sense to first area-average the TA and DIC values and *then* put them through the CO2SYS calculation ? It seems to me it would be more accurate to compute Revelle, pH, pCO2 etc for each surface location and or time and *then* do the area-weighted average of each metric. Perhaps over the range of values encountered the system is linear enough and this doesn't make much of a difference, but I'm not sure.

This study was meant to introduce the issue of alkalinity and DIC biases in ESMs, their implications for assessing model OAE experiments and to suggest some potential areas for model improvements. The suggested approach would certainly be a worthwhile follow-up study.

---

## Author Response (AR1)

**Point-by-point response to reviewer and editor comments below in blue.**

**Response to the editor:**

In the revised manuscript we now calculate (with CO2SYS) and discuss two metrics for assessing the models' response to alkalinity addition, i.e., OAE compared to GLODAP values: 1) the **initial** reduction in pCO2 and the 2) the change in ocean DIC **after assumed complete equilibration**, µCO2. The initial increase in pH and the pH increase after equilibration is calculated and discussed correspondingly.

As suggested by Reviewers 1 and 2 we have added the metric uptake efficiency, ηCO2, to our analysis. But unfortunately, in the response to the reviewers the new Figure 7 showed an incorrectly calculated uptake efficiency of ~0.63. This was rightly pointed out by the editor Dr. Sulpris. We have since corrected the computation of ηCO2 by using TA and pCO2 in CO2SYS to compute the ΔDIC directly (instead of using the computed Revelle factor to infer ΔDIC). Now, the values hover around the expected value of 0.83 (see new Figure 7):

[Figure]

*Figure 7 NEW: Carbonate system parameters were computed for all CMIP6 ESMs, the MMM (grey line) and the GLODAP data (black line) with the CO2SYS toolbox. The results are sorted by Revelle Factor in ascending order for all panels. Shown are the TA (a), the Revelle factor (b), the TA-DIC ratio (c), initial pCO2 (d), the difference in pCO2 after a 100 µmol kg-1 addition of TA (e), the uptake efficiency ηCO2 (f), the initial pH (g), the difference in pH for constant DIC (h), and the difference in pH for constant pCO2 (i). Light blue colors indicate the unperturbed mean state in the ESMs and GLODAP, dark blue colors the initial state after OAE and green colors the state after OAE and subsequent air-sea equilibration.*

We introduce the metric in the methods section:

*First, we evaluate the CO2SYS output fields Revelle Factor, pH, and $pCO_2$ (partial pressure of $CO_2$ in seawater) based on the CMIP6 ESMs against the values for the GLODAP data. In a second step, we assess the initial changes in surface pCO2 and pH after an addition of 100 µmol kg-1 TA (corresponds to 102.6 mmol m-3 TA) while keeping DIC constant. In a third step, we evaluate the CO2 uptake efficiency ($\eta CO_2$) (Renforth and Henderson, 2017, Tyka et al., 2022) and the pH difference at constant $pCO_2$ which simulates completed air-sea $CO_2$ equilibration. Note, that this calculation has an ocean-centric perspective as it assumes constant atmospheric CO2, which contradicts the motivation for ocean alkalinity enhancement to reduce atmospheric CO2, and thus will only be valid for small-scale applications. The uptake efficiency metric has been previously applied in ocean model simulations with constant and non-interactive atmospheric CO2 (Tyka et al., 2022; He and Tyka, 2023). We here follow this approach in our idealized calculations while acknowledging that atmospheric $CO_2$ would drop in emission-driven simulations (magnitude dependent on amount of alkalinity added; Ferrer Gonzalez et al., 2018; Lenton et al., 2018; Köhler, 2020), as in the real world, through feedbacks with the atmosphere and the land biosphere (Oschlies, 2008). The assumption of constant atmospheric $CO_2$ (and thus constant surface ocean $pCO_2$) was shown to overestimate oceanic $CO_2$ uptake by 2% on annual timescale, but by 25% on decadal timescale and further increasing on longer timescales (Oschlies, 2008).*

*The uptake efficiency, $\eta CO_2$, is the ratio of moles of CO2 absorbed to moles of added alkalinity and can also be expressed as the ratio of the partial pressure sensitivities of $pCO_2$ with respect to TA and to DIC (Tyka et al., 2022; Tyka et al., 2022):*

$$\eta CO_2 = \Delta DIC / \Delta TA \quad \#(5)$$

*For the uptake efficiency at constant $pCO_2$, the $\Delta DIC$ was also computed using CO2SYS, here with TA + 100 µmol kg$^{-1}$ and the initial $pCO_2$ as input parameters.*

And in the results section:

*We conclude that almost all ESMs might overestimate the initial additional pCO2 difference uptake in simulated OAE experiments. On the other hand, the CO2 uptake efficiency computed with constant pCO2 (equilibrated DIC) only differs by -0.1% to 1.1% (mean: 4.4%) from the GLODAP value, and the ESMs may thus represent equilibrium CO2 uptake rather robustly.*

And in the conclusions:

*The CO2 uptake efficiency factor, $\eta CO2$, relates changes in surface DIC to alkalinity input. We computed this metric here with constant pCO2 after alkalinity addition which suggests complete equilibration and neglects any reduction in atmospheric CO2 due to OAE. Studies suggest that the time scale and efficiency of the equilibration can differ immensely depending on the ocean region. He and Tyka (2023) found that after one year $\eta CO2$ varied between 0.2 and 0.85 and that after 10 years most locations showed an uptake fraction of 0.65–0.80. Jones et al. quantified the mean global air-sea equilibration timescale for CO2 at 4.4 months (range 0.5 to 24 months regionally). Bach et al. (2023) suggest a pragmatic time scale of 10 years for a 95% DIC equilibration after OAE measures. It is within this range of suggested equilibration time scales that the differences in simulated pCO2 change between ESMs are important.*

**Response to Reviewer 1:**

Overall, I found this to be an interesting, well-written and illuminating paper that I think will help spur improvements in model development. The separation of the TA biases into preformed TA, remineralization TA and CaCO3 TA is also very useful and points to concretely implementable improvements, especially in the treatment of CaCO3.

We thank the reviewer for their thoughtful und thorough review and helpful suggestions. We addressed the reviewer comments below in blue.

I recommend publication with some minor revisions (see below).

My two major comments are:

Line 254 & Figure 7

As the authors point out the biases in Revelle Factor are of great importance to mCDR. An additional metric of this that would be straightforward to add using CO2SYS and of great value to folks investigating the feasibility and cost of ocean alkalinity enhancement is the uptake efficiency factor (In our work we like to call this $\eta CO2 = \partial DIC/\partial Alk$ at constant pCO2, see https://doi.org/10.1039/D1EE01532J and https://bg.copernicus.org/articles/20/27/2023/). The metric simply indicates the number of moles of CO2 taken up per mol of Alkalinity added after full equilibration (for an infinitesimal increase in Alk) and is generally ~0.8 though it is quite dependent on location (see for example He et al., 2023, https://bg.copernicus.org/articles/20/27/2023/).

I think this number is very practical because it directly represents an efficiency loss going from some alkaline substance to actual CO2 drawdown and thus enters any cost estimates. Thus I would be very curious to see how model biases affect this metric, even if just expressed as a global average or surface average.

Thank you for the suggestion. We calculated uptake efficiency $\eta CO_2$ with CO2SYS and added the results to the new Figure 7 (see above)

The uptake efficiency ranges between ~0,825 and 0,835 for the models and GLODAP.

In response to Reviewer 2, we added TA in the same order to Figure 7. Models with a higher Revelle factor also have a higher uptake efficiency, $\eta CO2$, (R=0.98, Figure 7f) but overall, the CO2 uptake efficiency computed with constant pCO2 only differs by -0.1% to 1.1% (mean: 4.4%) from the GLODAP value, which means that the ESMs represent equilibrium CO2 uptake rather robustly.

L317ff and Figure 6, panel (d). There is clearly a large amount of difference in TA* between models and also in some models these biases are clearly depth-dependent while in others they are less so. This is one of the major insights of this paper. Not being familiar with the details of each of the models tested, I am very curious about whether there is any pattern or correlation between the sophistication of the CaCO3-cycle-model in each GCM and the amount and type of bias observed ?

E.g. The blue-ish models mostly overestimate TA at depth - do they have something in common in the way they treat the CaCO3 dissolution? Do any of these models treat the natural occurrence and distribution of CaCO3 sediments explicitly (see work by Sulpis et al and others for maps of this) ? or do they only account for precipitation and redissolution ? If not, then perhaps there is a spatial correlation between TA biases and occurence of CaCO3 sediments? I'd love to see more discussion of this phenomenon - it's very interesting! The discussion on L294-330 is in very general terms rather than looking at algorithm differences between the specific models that could explain the differences.

The paragraph describing the TA* bias has been reworked and amended:

*The bias in TA from calcium carbonate dissolution in the interior ocean (Figure 6d) can be in absolute terms comparable to or even larger than the bias in preformed TA. The MRI model and the GFDL models have a small negative bias in TA* in the order of ~10-20 mml m-3, relatively constant with depth. The MPI and NorESM models have a slight positive TA* bias in about the same order of magnitude, also relatively constant with depth, while the UKESM, the CNRM-ESM and the IPSL ESM exhibit TA* biases that increase with depth. The CNRM model has the largest TA* bias with about 100 mmol m-3 at 4,000 m depth. CNRM-ESM2-1 and IPSL-CM6A-LR, have in common that they contain the same ocean model (NEMO) and the same biogeochemical model (PISCESv2). Dissolution in PISCESv2 is treated explicitly and is dependent on the calcite saturation state and the sinking speed for PIC is depth-dependent, while for other models the sinking speed is constant. In two of the models (of Figure 6d), MRI-ESM2-0 and UKESM1-0-LL, CaCO3 is dissolved without a sediment, while the other models do have explicit sediment treatments where CaCO3 is buried or dissolved, either depend*

*on the calcite saturation state or a set rate (Planchat et al. 2023). A direct link of the treatment of CaCO3 at the bottom to the bias at depths is not obvious in this case.*

Figure Style comments:

As I was parsing the figures I felt some improvements in the plots could make better use of the space, aid visual parsing and generally make the paper even easier to follow. Please take these as suggestions, perhaps try them out and see if you like them.

Figure 1: Maybe expressing the MMM as a (say, dashed) line rather than an additional row would be more intuitive and allow visual comparison of each model vs the MMM ?

Figure 1: does the thickness of the GLODAP line have meaning (e.g. a standard error) or is it incidental ? If a standard error for the GLODAP measurement is known or can be computed it would be neat to use the thickness (using a semi-tranparent color) this way (unless the certainty is so high that it reduces to a thin line of course). I think this is important as a large GLODAP uncertainty could change or weaken the conclusions.

Figure 1, Line 264 As you note Alk and DIC are highly correlated, and they are compensating variables in the carbonate system, with respect to pH, pCO2 etc.

An alternative for the two panels in Figure 1 would be to plot both together as a scatter graph with DIC on the x-axis and Alk on the y axis (or vice versa). This way the exact same information is displayed but the extent of the correlation is visually immediately apparent as well. The scatter points could be labelled directly on the graph with a floating text for example. Error bars on each pint could indicate the variance of these values over the surface average.

*Thank you for the suggestion. In response, we replaced Figure 1 with a scatter plot of TA versus DIC, as we agree that this might be more intuitive. This figure also contains an error estimate for the global mean GLODAP data. While the MMM is almost within the estimated GLODAP range, we see that the difference for individual models can be quite large. We also see that TA and DIC biases are highly correlated, R = 0.99.*

[Figure]

*Figure 1 NEW: Global mean surface total alkalinity (TA) [mmol m⁻³] of the 14 CMIP6 models, the multi-model-mean (MMM), and GLODAP including its error estimate versus dissolved inorganic carbon (DIC) [mmol m⁻³].*

Figure 2&5: "Absolut error" → "Absolute error"

This has been corrected.

Figure 2&5: If the whitespace between globes could be reduced at all, that would make everything bigger and easier to parse (It's already tricky without looking at the PDF on a large screen).

Space between the subplots has been reduced as much as possible.

Figure 3: Ah, I see here there is a GLODAP error estimate. Great! Could this be added to Figure 1 also ?

Yes, this has been added to Figure 1. Thank you for the suggestion.

Figure 3&4: Style considerations: For a colour-blind person (like myself) it is nigh impossible to know which line is which, among similar shades/hues. I would recommend blending color with different dash/dot patterns to help with this. Perhaps the error (currently dashed lines) can be given simply by a transparent shaded area ?

We added linestyles to the profiles and indicated the GLODAP error estimate with shading.

[Figure]

Also, a lot of features of this graph occur in the upper 0.5km and are visually cramped in a very small area. For the same reason that most models have non-uniform vertical z slices, perhaps it would be possible to plot the vertical axis on a log scale or a mixed log-linear scale. Or split the graph into two linear regions, one for 0-500m and one 500-4000m ?

We would like to keep this figure as is since we were interested in how well the profiles overall match.

Figure 4(f) I'd adjust the x axis to not clip the values at shallow depth. Ah - I see all the X axes are coordinated. Hmmm. Not sure how to solve this. How low do the TA values go in the Arctic Ocean (last panel)? Perhaps one could plot all these as Deltas from GLODAP the same way that Figure 6 ? That might help with the dynamic range of the x axis (which IMHO does not necessarily have to be the same for each subpanel). I thought Figure 6 was very nice.

[Figure]

We adjusted the y-axis for the Arctic in Figure 4f and applied the linestyles here as in Figure 3 as well as added a line for MMM.

Figure 7: Again, using the thickness of the GLODAP vertical line to indicate variance would be neat.

The estimated error for the GLODAP values based on the TA and DIC GLODAP errors (see also new Figure 1) are very small for the computed parameters. We think the figure without the error would be clearer and kept the figure without error bars.

Figure 8: I found this figure rather difficult to parse. Because the three different variables have such different dynamic range on the %-scale, especially the second one (TA-DIC ratio) is virtually impossible to read off. Is this figure really necessary? I feel like most of the information content is already contained in Figure 7.

We will drop Figure 8 and instead refer to the percentage values listed in Supplement Table S2.

L87: Some other recent studies that would be worth including that also aim to be more realistic than the earlier large-scale uniform OAE simulations, i.e. near-coastal or ship-track-based releases or regional assessments.

https://doi.org/10.1002/2017EF000659
Model-Based Assessment of the CO2 Sequestration Potential of Coastal Ocean Alkalinization, Feng, Koeve, Keller, Orschlies 2017

https://doi.org/10.1029/2022EF002816
Simulated Impact of Ocean Alkalinity Enhancement on Atmospheric CO2 Removal in the Bering Sea,

Weng et al., 2022

https://doi.org/10.5194/bg-20-27-2023
Limits and CO2 equilibration of near-coast alkalinity enhancement, He and Tyka, 2023

https://doi.org/10.5194/egusphere-egu23-9305
Atmospheric CO2 removal by alkalinity enhancement in the North Sea, Liu et al. 2023

L87 has been appended to include most of the above suggested references and now reads:

*Now, more and more projects are underway or in planning that seek to apply more realistic scenarios for OAE e.g., in regional OAE applications (Butenschön et al. (2021), Wang et al. (2023) or coastal applications (Feng et al. (2017), He and Tyka (2023)), which is why a model evaluation is even more important.*

L149: Since the carbonate system isn't linear wrt TA and DIC, does it make sense to first area-average the TA and DIC values and *then* put them through the CO2SYS calculation ? It seems to me it would be more accurate to compute Revelle, pH, pCO2 etc for each surface location and or time and *then* do the area-weighted average of each metric. Perhaps over the range of values encountered the system is linear enough and this doesn't make much of a difference, but I'm not sure.

This study was meant to introduce the issue of alkalinity and DIC biases in ESMs, their implications for assessing model OAE experiments in an idealized approach and to suggest some potential areas for model improvements. The suggested approach would certainly be a worthwhile follow-up study, also to look at regional differences with regard to the here described biases.

**Responses to reviewer 2:**

We thank Alban Planchat for his thoughtful und thorough review and helpful suggestions. We addressed the reviewer comments below in blue.

Review of Hinrichs et al. (2023)
Overall comment:
In overall, this work is interesting and innovative in its approach to bias assessment in OAE modelling studies using ESMs. Indeed, the study pushes a CMIP6 bias analysis (essentially focused on alkalinity, but also on DIC) towards a concrete, but idealized, CDR experiment. The approach used to decompose the alkalinity bias is broadly consistent with Planchat et al. (2023), thus reinforcing the message to the
modelling community around the representation of alkalinity and the carbonate pump in ESMS. The writing of the manuscript could nevertheless be partly improved, in particular some parts could be restructured to gain clarity. Many technical corrections should be made.

Specific comments
**A. Main comments**
1. In the decomposition of the alkalinity bias, it is repeatedly mentioned that "preformed alkalinity" refers to physical biases for the ocean, but it also contains biases of both organic matter and CaCO3 production (e.g., l. 17 ; l. 121-125 ; l. 218 ; l. 291 ; l. 294)

We amended L 17 with:

*The decomposition of the global mean alkalinity biases into contributions from physical processes (preformed alkalinity) which includes i) the physical redistribution of biased alkalinity originating from the soft tissue und carbonates pumps, ii) remineralization, and iii) carbonate formation and dissolution showed that the bias stemming from the physical redistribution of alkalinity is dominant.*

We amended the method part L121-125 on preformed alkalinity with:

*Preformed Alkalinity also refers to the physical redistribution of alkalinity biases stemming originally from soft tissue and carbonate pumps and the upwelling of water masses with biased alkalinity.*

L221 was amended with the sentence:
*The first part, preformed alkalinity, includes the advection and upwelling of already biased water masses.*

2. The direct link made with the CO2 uptake in the OAE idealized experiment is confusing for me (e.g., 274, l. 327): a relative pCO2 difference should not directly be a relative CO2 uptake difference, or should it?
According to me, we have:
$dDIC/DIC = 1/Re \cdot dpCO/pCO$
An ESM with a greater relative variation of pCO2 resulting from alkalinity addition, also has a greater Revelle factor. So, the product of the right member of the above equation should at least partly compensate. To access a first approximation of the CO2 uptake associated with alkalinity addition, the relative variation of DIC should be expressed in function of the relative variation of alkalinity. I think that with what is currently offered, it is possible to assess the bias resulting from OAE in ESMs concerning acidification, but I'm confused by the direct link made with the CO 2
uptake.

Thank you for bringing up this point. We have now added the parameter uptake efficiency ($\eta CO_2$) to our back of the envelope calculation. As you said, the uptake efficiency depends on the ratio of $pCO_2$ change over pCO2 state with respect to total DIC over alkalinity change (or addition) and the Revelle factor:

$\eta CO_2 = \partial DIC/\partial Alk$ at constant $pCO_2$,

*As you said, the inherent model states of pCO2, TA and DIC lead to slightly different efficiencies for a constant alkalinity addition. On your suggestion, we added the TA plot into Figure 7 as well (see above). And discuss the correlation between the parameters in the results section:*

*The global mean Revelle Factor from the CO2SYS computation for the GLODAP dataset is with 10.19 the third lowest in our compilation and thus almost all ESMs have a higher Revelle Factor than the GLODAP data, ranging from 10.18 to 10.54 (Figure 7b). The Revelle factor is anti-correlated to the average TA-DIC-ratio (R=-0.99, Figure 7c). Also, the order of surface pH (R=-0.96, Figure 7g) and pCO2 (R=0.97, Figure 7d) values corresponds largely to each model's rank in Revelle Factor (and thus also with TA-DIC-ratio). Models with a higher Revelle factor than GLODAP have a lower buffer capacity, which leads to already higher pCO2 values (290 to 314 µatm) and lower pH (8.12 to 8.17) than in GLODAP (pCO2: 292 µatm, pH: 8.16). Those models also show a greater initial reduction in surface ocean pCO2 for the hypothetical addition of 100 µmol kg-1 of TA (R=-0.99, Figure 7e) than GLODAP (-92 µatm), ranging from a 91 µatm to a 104 µatm decrease in pCO2. Models with a higher Revelle factor also have a higher uptake efficiency, ηCO2, (R=0.98, Figure 7f). The initial change in pH after alkalinity addition (Figure 7h) is about an order of magnitude larger than the change in pH after complete air-sea equilibration at constant atmospheric CO2 (Figure 7i). The respective changes in pH (unequilibrated / equilibrated at constant atm CO2) have a higher correlation to TA (R=-0.92, R=-0.99) than to the Revelle factor (R=0.83, R=0.63).*

**B. Minor comments**
1)
The second part of the abstract could be improved and made more readable. In particular, I do not understand: "We find that the degree of compensation of DIC and alkalinity biases at the surface is more important for the marine CO2 uptake capacity than the alkalinity biases themselves."

We adapted the paragraph, which now reads:

*The relative biases of alkalinity versus DIC at the surface affect the Revelle factor and therefore the initial pCO2 reduction after alkalinity addition. The global mean surface alkalinity bias relative to GLODAPv2 in the different models ranges from -85 mmol m-3 (-3.6%) to +50 mmol m-3 (+2.1%) (mean: -25 mmol m-3 or -1.1%). For DIC the relative bias ranges from -55 mmol m-3 (-2.6%) to 53 mmol m-3 (+2.5%) (mean: -13 mmol m-3 or -0.6%). All but two of the CMIP6 models evaluated here overestimate the Revelle factor at the surface by up to 3.4% and thus overestimate the initial pCO2 reduction after alkalinity addition by up to 13%. The uptake efficiency, ηCO2, then takes into account that a higher Revelle factor and a higher initial pCO2 reduction after alkalinity addition and equilibration mostly compensate, so that resulting DIC differences in the models are small (-0.1% to 1.1%). The overestimation of the initial pCO2 reduction has to be taken into account when reporting on efficiencies of ocean alkalinity enhancement experiments using CMIP6 models especially as long as the CO2 equilibrium is not reached.*

2)
(l. 25-26); and l. 27-29 are too heavy,
Broke this up into two sentences.

perhaps misplaced, and with a unit error (mmol/kg instead of mmol/m3 as l. 209-211).
Thanks for pointing this out, it should be mmol/m3. This has been corrected.

3)
For the sake of clarity and consistency throughout the manuscript, I would suggest that alkalinity and DIC should always be shared in mmol/m3 rather than mixing mmol/m3 and umol/kg.
Changed to mmol/m3 in Figure 1 and text where applicable.

It would simply be necessary to clarify how the conversions are done for GLODAPv2 and for using CO2SYS in the methodology (see l. 28-29, l. 149-152, Fig. 1 etc.).
For the model evaluation the GLODAP TA and DIC data is converted from units of µmol kg$^{-1}$ to mmol m$^{-3}$ using the potential density computed from GLODAP salinity and temperature data. We added this to the methods section.

4)
Table 1: It would be great also to share the ensemble member for each ESM, rather than writing "the first available ensemble member" l. 109, which is slightly confusing.
The ensemble member has been added to Table 1.

5)
The marine biogeochemical model of CNRM-ESM2-1 (not CNRM-ESM-2-1) is PISCESv2-gas. The one
for IPSL-CM6A-LR is PISCESv2. The one for NorESM2-LM/MM is iHAMOCC.
This has been corrected.

Why do you precise the grid only for a few ocean models? If you wish to precise it, add it in the "ocean horizontal resolution" column and just precise maybe whether it is a tripolar grid or not.
Added information about tripolar grids to grid column.
6)
l. 109-119: These lines would benefit from restructuring.
The TA* method: are there limitations to this decomposition of the biases?
Koeve et al. (2014) discuss the limitations as follows: although the computation of TA* according to equations (2) to (4) reproduces tracer-based simulated TA* robustly in most of the global ocean, higher uncertainties occur in the Atlantic and in the 500m to a 1,000m layer in the Pacific and Indian ocean.
In our study we focus on the TA* results for the global mean ocean.

Another caveat that was mentioned by Koeve et al. is related to the issue that AOU overestimates true oxygen utilization by 20–25 %. Hence TAr computed from AOU is probably also overestimated by this percentage. TAr is small and, in our study, we focus on the implication of TA* biases in ESMs and potential remedies for these biases.
This short discussion of the above-mentioned caveats of the Koeve method was added to the discussion part.

What are the advantages of it rather than the one suggested in Sarmiento and Gruber (2006)? Not using DIC?
In some ways the methods are similar. Both are partly based on Feely et al. (2002). E.g., the TA_r computed in Koeve et al. (2014) is similar to $\delta C_{soft}$. Koeve then aims to further separate the influence of ocean physics from the impact of carbonate dissolution. Although, as the reviewer correctly points out the physical distribution of TA includes the distribution of TA biases from the biological and the carbonate pump as well. Another difference is that preformed alkalinity is computed for each grid point and therefore is resolved spatially (but here shown as global profile only).

We added a short discussion on these methods in the revised manuscript.
7)
Eq. (3) and (4): why do you use phosphate for PO in Eq. (3) and then nitrate in Eq. (4) for TAr rather phosphate (or nitrate) in both cases?
We were following the equations proposed by Koeve et al. (2014) where using PO (NO) is proposed for computing TA_0 interchangeably and no3 for computing TA_r in the manuscript.

Here we recomputed TA_0 and the TA* using NO not PO, below are both ways to compare.

TA_0 computed with PO:

[Figure]

TA_0 computed with NO:

[Figure]

The difference in TA* using PO versus NO is really small, since the bias profiles in PO4 and NO3 are very similar for most models and the regression coefficient for the PO (NO) part is very small. Most 'weight' in the multilinear regression is on the salinity.

8)
l. 139, you explicitly mention the fact that TA* "is computed as residual after rearranging Eq. (2)". This is questioning, since it is finally on the dissolution of CaCO3 that we have the more certainty over its affects on alkalinity (+2 eq for 1 eq of CaCO3 dissolved). On the contrary, the effect of remineralisation, for example, is more complicated to extract from ESMs because of the N-reactions whether or not
they are taken into account in the biogeochemical models. It is therefore perhaps a pity not to take advantage of the ease with which the carbonate pump can be used in terms of its effect on alkalinity with this method.

Yes, as the reviewer points out, there are uncertainties related to N-reactions, i.e., of reaction of PON to ammonium to nitrate. However, on the one hand, this is a fast reaction and on the other hand most models don't simulate ammonium explicitly. We thus think that the uncertainty that goes with neglecting the effect from N-reactions is small at present. Koeve et al. (2014) states that the "global mean ratio of the alkalinity effect stemming from organic matter remineralization to TA background is only 0.005. Contributions from N2-fixation, denitrification, sulfate reduction, and shelf alkalinity fluxes are even smaller or of local importance only.
Koeve et al. (2014) also tested the a posteriori TA* method used here against explicitly modeled tracer of TA0, TAr and TA* with good results on a global scale.

9)
l. 156-157: Could you explain why you have chosen these values rather than the mean surface ESM values for instance?
We chose to apply the same typical values for all models. We now did a test and applied the actual MMM surface salinity (34.5), temperature (18.4) and PO (0.5) values for the calculation and found that the differences in the results are very, very small.

| Model | % difference in PCO2 after alkalinity addition using S=34, T=15, PO=2 | % difference in PCO2 after alkalinity addition using S=34.5, T=18.4, PO=0.5 |
|---|---|---|
| 'MRI-ESM2-0' | -1.02 | -0.99 |
| 'ACCESS-ESM1-5' | 0.16 | 0.14 |
| 'CESM2' | 2.53 | 2.47 |
| 'GFDL-ESM4' | 1.54 | 1.54 |
| 'CESM2-WACCM' | 2.83 | 2.77 |
| 'NorESM2-LM' | 3.55 | 3.47 |
| 'MMM' | 4.94 | 4.87 |
| 'NorESM2-MM' | 4.22 | 4.21 |
| 'GFDL-CM4' | 5.83 | 5.73 |
| 'CNRM-ESM2-1' | 6.06 | 5.96 |
| 'UKESM1-0-LL' | 6.99 | 6.88 |
| 'IPSL-CM6A-LR' | 7.70 | 7.57 |
| 'CanESM5' | 6.90 | 6.83 |
| 'MPI-ESM1-2-HR' | 10.63 | 10.44 |
| 'MPI-ESM1-2-LR' | 12.98 | 12.76 |

10)
l. 166-168: This sentence should be part of the methodology.
Sentence was moved to methods part.

11)
l. 190 and Fig. 3b: The very high TAn value observed at the ocean surface for GFDL-ESM4 is linked to very low salinity values (even possibly null) in the Baltic Sea if I remember well. You might be able to neglect this closed sea for this ESM when averaging to avoid this issue. You could also use a salinity threshold maybe.
Thank you for the suggestion, this issue has been fixed by using a salinity threshold of 10, which we noted in the methods section.

12)
Fig. 3 and 4: I would combine both figures so that the first panel is not repeated.
We would like to keep the comparison of TA and TAn separate from the comparison of global to regional profiles.

13)
l. 201: It could be mentioned that the bias observed for CNRM, IPSL and UK is associated with the ocean model NEMO.
Reads now:
*Most models reproduce this pattern, while the CNRM, IPSL and UK ESMs simulate a strong increase of TA below about 2,000 m depth (Figure 4b). Those three ESMs have a NEMO ocean model in common.*
14)
l. 211: "TA biases likely lead DIC biases, as DIC can adjust through gas-exchange of CO2 ". I suggest to add at the end "to maintain a surface chemical equilibrium with the atmospheric CO2 concentration".
The suggested sentence has been added.
Besides, this aspect could be slightly more highlighted to understand why you decompose the alkalinity biases and not the DIC biases in addition, whereas you consider both surface alkalinity and DIC in the following with CO2SYS.
We focused on TA because of its conservative nature and because we came from the background of alkalinity enhancement studies. One result then became that DIC biases can act in a compensating way.

15)
l. 221-230: This paragraph should be rewritten.
It should be readable and meaningful independently of Fig. 6 (avoiding "are shown in Figure 6" instance).
The paragraph has been rewritten:

*The decomposition of the TA biases (Figure 6a) shows that in the upper 1 km most of the models' alkalinity biases are due to their preformed TA (Figure 6b). Per definition, models with a negative surface TA bias have a negative bias in preformed TA. Below about 1,000 m depth $TA^0$ stays constant with depth. TA biases from the representation of organic matter remineralization processes are in the order of 5 to 10 mmol $m^{-3}$ and play only a negligible role in absolute terms (Figure 6c). The bias in TA from calcium carbonate dissolution in the interior ocean (Figure 6d) can be in absolute terms comparable to or even larger than the bias in preformed TA. The MRI model and the GFDL models have a small negative bias in TA\* in the order of ~10-20 mml $m^{-3}$, relatively constant with depth. The MPI and NorESM models have a slight positive TA\* bias in about the same order of magnitude, also relatively constant with depth, while the UKESM, the CNRM-ESM and the IPSL ESM exhibit TA\* biases that increase with depth. The CNRM model has the largest TA\* bias with about 100 mmol $m^{-3}$ at 4,000 m depth. CNRM-ESM2-1 and IPSL-CM6A-LR, have in common that they contain the same ocean model (NEMO) and the same biogeochemical model (PISCESv2). Dissolution in PISCESv2 is treated explicitly and is dependent on the calcite saturation state and the sinking speed for PIC is depth-dependent, while for other models the sinking speed is constant. In two of the models (of Figure 6d), MRI-ESM2-0 and UKESM1-0-LL, $CaCO_3$ is dissolved without a sediment, while the other models do have explicit sediment treatments where $CaCO_3$ is buried or dissolved, either depend on the calcite saturation state or a set rate (Planchat et al. 2023). A direct link of the treatment of $CaCO_3$ at the bottom to the bias at depths is not obvious in this case.*

The first sentences are too repetitive compared to what was already described in the previous sections.
Removed the repetitive sentence.
l. 226-228: This sentence is out of topic as you are talking about biases in this section. It would be great also to share some values.
The sentence has been removed and some values are added, see new paragraph above.
16)
Fig. 6: The addition of MMM would be meaningful on these plots.
The MMM has been added to Figure 6.

[Figure]

17)
l. 237: I am not convinced of the veracity of the causal sequence (cf. "thus"). Could you detail it?
Biases in simulated surface TA and surface DIC have implications for the individual models' efficiency of OAE in terms of Revelle factor and change in pCO2 and thus in the marine CO2 uptake capacity. This is now shown in the newly added μCO2 metric in Figure 7.

18)
l. 240-242: "All panels are sorted by Revelle factor in ascending order." This is enough in the legend.
Sentence was removed.
19)
"The Revelle factor ... at the ocean surface." should be part of the Introduction.
Removed here.
20)
Fig. 7: Could you set "MMM" in bold so that we can easily spot it?
Changed the figure to also have vertical MMM bars. Please see above.

21)
I also suggest to keep the x-axis increasing in the last panel, to avoid confusions.
Okay, this has been changed.
22)
l. 260-261: Too repetitive with what was already said for Fig. 1.
23)
l. 262: It is less than 13.0 % in Fig. 8, I think: t should be about 10.7 %.
Figure 8 has now been removed following the suggestion by Reviewer 1. In the initial figure though, the model with largest deviation, MPI-ESM-LR, of ~13% was missing. Thanks for pointing this out.

All values are listed in Table S2. The maximum value is 12.98 %.

24)

l. 263: It would be great to have also the TA bias in Fig. 7 with the ESMs ordered in the same way.

Thank you for the suggestion. TA has been added to Figure 7, which helps to illustrate our findings.

25)

Fig. 8: Could you set "MMM" in bold so that we can easily spot it?

Could you adapt the y-axis scale to avoid the blank space to the top?

Figure 8 has been removed in response to a comment by reviewer 1.

26)

Discussion and conclusions: In overall, this section could be improved in terms of organization and content. There are some repetitions and it is sometimes difficult to follow you.

In particular, l. 310-323 are quite messy for instance and the points overlap each other sometimes.

We sorted the suggestions for improvement into points that relate to tuning and points that relate to implementing or expanding parametrizations.

In the first point, are you pointing towards including aragonite as well as calcite?

The first point was a suggestion to tune the calcification rate.

What would be the effect of reducing the calcification rate in the Southern Ocean, where deep waters are upwelled?

Biases related to primary production and the calcification rate would be distributed, relatively speaking, equally at the ocean surface and in the upwelling regions of the Southern Ocean there is very little calcification anyway.

For almost all ESMs the TA bias is the same sign in the SO as in most of the global ocean (see Figure 2) so that tuning the calcification rate would have the same directional impact.

Hopefully, this answers the question, otherwise we are not quite sure what specifically the reviewer is asking.

27)

l. 320: are you talking about calcite dissolution (effectively sometimes explicitly modelled) or calcite production (always implicitly modelled in that case)?

The carbon cycle formulation could be expanded to also simulate aragonite, i.e. including both aragonite formation and dissolution. Indeed, it is more important to separate aragonite and calcite for dissolution than for CaCO3 formation.

28)

Another important point, which is not mentioned, is the burial and dissolution at the seafloor.

We added a bullet point to the list :

•    *The representation of CaCO3 treatment at the bottom-sediment interface (dissolution, sedimentation, sediment weathering) is important for the total alkalinity budget and also for upper ocean alkalinity especially in more shallow regions where alkalinity-enriched waters (through dissolution) can recirculate to the upper ocean more quickly (Gehlen et al., 2008).*

29)

Fig. S2: It is not compared to the GLODAP climatology here, as opposed to Fig. 6.

Figure S2 shows the results from the TA* method for the models and GLODAP side-by-side. GLODAP is the thick black line.

30)

Table S1: Not useful, I think.

We would like to keep it, just in case.

31)

There might be some issues in the References. There is at least one for Planchat et al., 2023, since some authors are missing.

Corrected the reference.

**Technical corrections**

All technical corrections have been addressed in the revised manuscript. Thanks again for your thorough review!

*The following remarks are valid for the whole manuscript, and only a few occurrences will be mentioned*

*below:*

- There is a regular lack of punctuation, mainly commas.

- Beware of citations: a number of references appear with double brackets in the text.

- In intercomparison studies of ESMs, it may be more accurate and preferable to refer to ESMs throughout the study rather than models.
- If you decide not to put a space before '%', do so everywhere.
- I suggest to write 'TA* method' instead of 'TA* Method'
- The p in $p$CO2 should be in italic
- When the abbreviation/acronym has been defined, it can then be used directly (especially in the legend of the figures)
- Try to keep the text independent of the figures and mention the figures in parentheses instead of within the text directly
- Write "Revelle factor" instead of "Revelle Factor"
- Always write "TA-to-DIC-ratio" instead of mixing the way you mention it
l. 4: 1 should in exponent
l. 9-15: commas are missing
l. 10: … on alkalinity …
l. 10: That is why, in the search for …
l. 11: … Ocean Alkalinity Enhancement (OAE) …
l. 12: … exists on how …
l. 13: … 14 CMIP6 Earth system models (ESMs) …
l. 17: … shows …
l. 36: … overestimate it in the …
l. 40-42: commas are missing
l. 45: … species composing the so-called …
l. 45-47: Suggestion: The oceanic uptake of anthropogenic carbon leads to an increase in aqueous CO2
and thus DIC. By changing the chemical equilibria between the carbonate species, this results in ocean
acidification with a decrease in pH.
l. 48: … over acids (proton donors) …
l. 48: … role in the partitioning of the DIC pool, especially in the form …
l. 49-56: Very nice paragraph, but the first sentence could potentially be improved. Suggestion: The carbonate system is key in driving the seawater ability to resist a change in its chemistry, also called its
buffering capacity. In particular, the Revelle factor is the sensitivity of $p$CO2 to changes in DIC. A low Revelle factor indicating a high buffering capacity and vice versa.
l. 55: … the potential CO2 uptake …
l. 68: … much research has focused on …
l. 72: … (OAE; Köhler et al., 2013 …
l. 81: … set-ups (e.g., Ilyina et al., 2013; … ; Burt et al., 2021).
l. 86-87: … underway, or in planning, seeking to apply … for OAE (e.g., Butenschön et al., 2021), highlighting the importance and urgency of a robust ESM evaluation.
l. 93: They report an improvement in …
l. 109: "the first available ensemble member" is confusing and not precise.
l.111: (CDO; Schulzweida, 2022)
Eq. (1): Precise that S is the salinity
l. 115-119: Suggestion: … against gridded observational products: (i) TA, DIC and pH from the GLODAPv2.2016b Mapped Climatology (GLODAP in the following; Lauvset et al., 2016); (ii) oxygen and nutrients from the World Ocean Atlas 2018 dataset (WOA; Garcia H.E., 2019) and GLODAP; and (iii) salinity and temperature from the Polar science center Hydrographic Climatology (PHC3.0; Steele et al., 2001) and WOA. For the evaluation of global mean vertical profiles, the model data are interpolated onto the same 33 vertical levels used in the GLODAP climatology.
l. 125: … calcium carbonate (CaCO3) dissolution (TA*)
l. 128 : potential temperature
l. 129 : if NO is mentioned, then it should be defined to explicitly mention the difference with PO.
l. 131, 136: could you add references for the ratios $r_{(…)}$ that you consider?
l. 131-132: "onto upper ocean TA values" and then "interior ocean" is a bit contradictory
l. 140: … to 10 of the 14 CMIP6 ESMs (… NorESM2-MM and UKESM1-0-LL)
l. 143: 2.3. Theoretical model sensitivity to OAE
l. 145: OAE. Thus, …
l. 147-148: Suggestion: this toolbox, from the combination of two of the carbonate system variables, compute the entire ocean CO2 system.
l. 149-152: Suggestion: … 1,026 kg/m3 (Fig. 1). First, we evaluate the CO2SYS output fields : Revelle

factor, pH and $p$CO2 (partial pressure of CO2 in seawater) based on the CMIP6 outputs. Then, we assess
the changes…

l. 155 … (MMM). Black vertical … value. (b) Same …

l. 156: potential temperature

l. 164: … at the surface (Fig. 1a and 2)

l. 165: (Fig. 1a and 2)

l. 170: … in NorESM-LM/MM and CanESM5; Fig. S1).

Fig. 2: Panel 1, title: GLODAP

Panel 2, title: GLODAP error; colorbar title: absolute error.

General colorbar title: Surface TA bias [mmol/m3]

Legend: Surface distribution of TA in GLODAP (top left), its error estimate (top center) and the CMIP6 multi-model-mean (MMM) bias (top right), as well as the respective biases of the ESMs.

Same comments for Fig. S1, where TAn should be directly mentioned.

Same comments for Fig. 5.

Try to homogenize the legends too

l. 177: … the increase of TA with depth deeper …

l. 179: … TA overall. This indicates that their global inventory of TA is respectively too low and too high …

l. 182: … in the GFDL ESMs …

l. 185-186 1,024; 1,028; 1,035

l. 187: … at depth, or vice versa, …

l. 189: … in the upper ocean. The surface minima …

Fig. 3: Could the black and grey lines have no transparency and be to the forefront so that we can clearly see both of them?

Legend: … and TAn, (b) …

l. 199: … by the ESMs, referring to circulation biases.

l. 200: … Ocean), between … depth, …

l. 208: … and local distribution …

l. 216: 3.2 Decomposition of the vertical alkalinity biases

l. 219: … (CaCO3) dissolution and …

Fig. 6: subplot titles: I suggest to simply write "TA/TA0/TAr/TA* bias"

l. 235: Impact of biases on OAE efficiency

l. 236: … surface TA and DIC …

l. 238: … was conducted, using surface TA and DIC, to calculate the full carbonate system (see Methods
and Fig. 1).

l. 248: … GLODAP data, ranging from …

Fig. 7: … CO2SYS toolbox. The results …

l. 269: … OAE idealized experiment shows that 12 out of 14 ESMs …

l. 278-279: … mmol/m3; i.e., YY %)

l. 294-295: In the sub-surface and the deep ocean, biases in TA are also driven by the CaCO3 dissolution, while contributions from remineralization of organic matter are negligible